# Deep learning-based object detection on LiDAR-derived hillshade images: Insights into grain size distribution and longitudinal sorting of debris flows

Paul E. Schmid[1], Jacob Hirschberg[1,2], Raffaele Spielmann[1,2], and Jordan Aaron[1,2]

[1]Chair of Engineering Geology, Department of Earth and Planetary Sciences, ETH Zürich, Zürich, Switzerland
[2]Swiss Federal Institute for Forest, Snow, and Landscape Research WSL, Birmensdorf, Switzerland

**Correspondence:** Jacob Hirschberg (jacob.hirschberg@eaps.ethz.ch)

**Abstract.** Debris flows are hazardous natural phenomena characterized by rapid movements of sediment-water mixtures in steep channels, posing significant risks to life and infrastructure. Better understanding and managing these hazards requires new methods to collect and process high-resolution data. This study introduces a novel method that leverages hillshade images derived from a high temporal resolution LiDAR scanner and deep learning-based object detection models to analyze debris-flow dynamics. By transforming 3D point clouds into hillshade projections, the method enables efficient detection and tracking of key flow features, including boulders, rolling boulders, surge waves, and woody debris, independent of ambient light conditions. Outputs include object velocities, sizes, and tracks, offering high-resolution insights into debris-flow phenomena such as longitudinal sorting. Six state-of-the-art object detection models were evaluated, with YOLOv11 achieving the best balance of precision, recall, and processing speed. We used the framework to calculate dynamic grain size distributions and found that the median grain size decreased continuously throughout the event. The proposed framework is scalable, significantly reduces processing time compared to manual analysis, and sets the foundation for real-time monitoring and early-warning of debris flows across diverse locations and conditions.

## 1 Introduction

Debris flows are rapid mass movements (>5 m/s) characterized by a mixture of sediment and water, typically occurring in steeply inclined channels (Hungr et al., 2014). Due to their high velocity and surging behavior, they cause thousands of fatalities per year and significantly damage infrastructure, agricultural land, and property (Dowling and Santi, 2014; Prakash et al., 2024; Hilker et al., 2009). The risk posed by debris flows is steadily increasing as development pressures drive construction into areas prone to these hazards (Jakob and Hungr, 2005). Furthermore, debris-flow activity in high alpine areas has increased in the last decades and related hazards are generally expected to be affected by climate change (Hirschberg et al., 2021b; Kaitna et al., 2023; Jacquemart et al., 2024). A better understanding of fundamental debris-flow mechanisms is required for future management of these changing boundary conditions. However, we currently lack in-situ measurements of individual features from multiple stations along a debris-flow channel. This has limited our understanding of the interactions among individual

features (boulders, surge waves, woody debris) and the surrounding liquefied slurry, which is essential when assessing impacts on mitigation structures or buildings (Hungr et al., 1984; Song et al., 2018).

Debris-flows feature a characteristic bouldery front, and theoretical and experimental studies suggest that it is generated by longitudinal sorting of the constituent material, whereby coarse particles are preferentially moved to the front due to the vertical velocity profile of the flow (Gray, 2018). Spielmann et al. (2025) recently proposed an extension of this mechanism for the case where particle size approaches the flow depth, and proposed that cross-channel (as opposed to vertical) velocity profiles can also transport large particles. Furthermore, as the front progresses more slowly than the liquified slurry (Aaron et al., 2023; Spielmann et al., 2025), large boulders may be deflected or overridden within the flow (Pouliquen et al., 1997; Thornton and Gray, 2008). However, very few in-situ measurements of these processes have been made, which limits our ability to understand and predict the important influence of sorting on debris-flow motion.

Another important phenomenon associated with debris-flows is the occurrence of surges (Iverson, 1997; Hungr et al., 2014). These surges often show a wave-like appearance and move at a higher velocity than the main flow, which in turn increases

impact pressures and enhances their destructive power (Edwards and Gray, 2015). The causes of these surges may include longitudinal sorting of materials (Iverson et al., 2010) or temporary accumulation in gentler channel slopes, followed by sudden release (Kean et al., 2013). They are also frequently explained by mechanisms similar to the formation of roll waves in open channel flows of water (Zanuttigh and Lamberti, 2007; Arai et al., 2013; Aaron et al., 2025). As outlined above, the internal processes, the dynamics of individual objects, and the development of surges remain poorly understood, requiring monitoring

at higher spatial and temporal resolutions.

Debris flows occur repeatedly within established catchments and known channels (Hungr et al., 2014), which allows for in-situ monitoring of key flow variables. However, the spatial- and temporal resolution of past in-situ measurements has limited our understanding of the interactions between debris-flow features, channel geometry, and flow velocity. Different monitoring systems can capture various aspects of debris-flow dynamics (cf. Hürlimann et al., 2019), but they are often unable to detect a

high number of debris-flow features in both day and night conditions. Recently, we utilized data from multiple high-frequency LiDAR sensors installed at the Illgraben catchment (Valais, Switzerland) to measure key debris-flow parameters (Aaron et al., 2023; Spielmann et al., 2024; Åberg et al., 2024; Hirschberg et al., 2025; Aaron et al., 2025; Spielmann et al., 2025). However, processing the substantial volume of 3D point-cloud data generated during these events (1-2 TB per event) is both time-consuming and labor-intensive. Therefore, an automated method is required to track a greater number of objects more efficiently

than is feasible for a human operator.

Convolutional neural networks (CNNs) offer a potential solution for object detection by automating the identification of targets and reducing dependence on manual annotation. While deep learning models like PointNet (Qi et al., 2017) and PIXOR (Yang et al., 2019) have demonstrated effectiveness in processing point-cloud data, challenges remain due to the high dimensionality, sparsity, and unstructured nature of such data, which demand significant computational resources and extensive an-

notation efforts (Fernandes et al., 2021). Aaron et al. (2023) addressed this challenge by taking advantage of the spatial surface structure of debris flows to generate hillshade images from 3D point clouds and applying a particle image velocimetry (PIV) method to the image. Additionally, a LiDAR-camera fusion method was introduced, that leverages optical RGB images from

a stationary camera to detect individual features within the debris flow. By combining optical images with point-cloud data, valuable 3D information was extracted. Building on this approach, Hirschberg et al. (2025) refined the method and evaluated the generalizability of two object detectors. This resulted in high detection accuracies and the development of a multi-domain detector, replacing the need for separate detectors configured to individual locations.

In this study, we introduce a new method to detect and track debris-flow features based solely on LiDAR data, thereby being independent of ambient light, and enabling the estimation of object velocities, sizes, and trajectories for thousands of debris-flow features. We analyze data captured by multiple high-resolution LiDAR sensors installed at various locations along the fan of the Illgraben during a debris flow, allowing us to study debris-flow evolution in time and space. Visualizing point clouds as hillshades simplified the processing and allowed us to utilize modern CNN object detection algorithms, originally developed for optical images. Here, we analyze and compare three versions of YOLO (v5, v8, and v11), along with RT-DETR, RetinaNet, and Faster-RCNN for detection and tracking of debris-flow features. This automated approach establishes a foundation for future investigations and early-warning of debris-flow events across diverse locations and conditions.

## 2   Study site

### 2.1   Illgraben catchment

The study site is the Illgraben (46.27° N, 7.61° E), a torrent system located in the municipality of Leuk, canton of Valais, Switzerland. The Illgraben is one of the most active debris-flow catchments in the Alps, where hillslope processes deposit sediments in the channel system. These sediments are then mobilized after intense rainfall, developing into debris flows, which can reach the Rhône river (Fig. 1a McArdell and Sartori, 2021). To mitigate damage to the nearby village of Susten, concrete check dams (CDs) have been built along the channel (Lichtenhahn, 1971).

The catchment is underlain by fractured bedrock forming an anticline within the Penninic nappe stack (Schlunegger et al., 2009). The Illgraben channel follows the southwest-trending axial plane of this structure, incising Triassic schists and dolo-breccias that provide the fine-grained matrix of the debris-flow deposits. Steep limestone cliffs (>50°) on the northwest side primarily generate rockfalls, whereas quartzite slopes on the southeast side (30–40°) produce frequent debris flows that supply quartzite clasts and fine material to the channel (Schlunegger et al., 2009). Further information on the geological and geomorphological setting of the Illgraben catchment is provided, for example, by Gabus et al. (2008); Schlunegger et al. (2009) and McArdell and Sartori (2021).

### 2.2   Debris-flow monitoring

The Illgraben has been monitored by the Swiss Federal Institute for Forest, Snow and Landscape Research (WSL) since June 2000 (Hürlimann et al., 2003, 2019), with several debris flows recorded annually (Hürlimann et al., 2003; Hirschberg et al., 2021a). Over the years, researchers have deployed a range of sensors to measure diverse debris-flow properties, including front velocity, flow depth, bulk density, normal and shear forces, and erosion depth (e.g., Badoux et al., 2009; Bennett et al.,

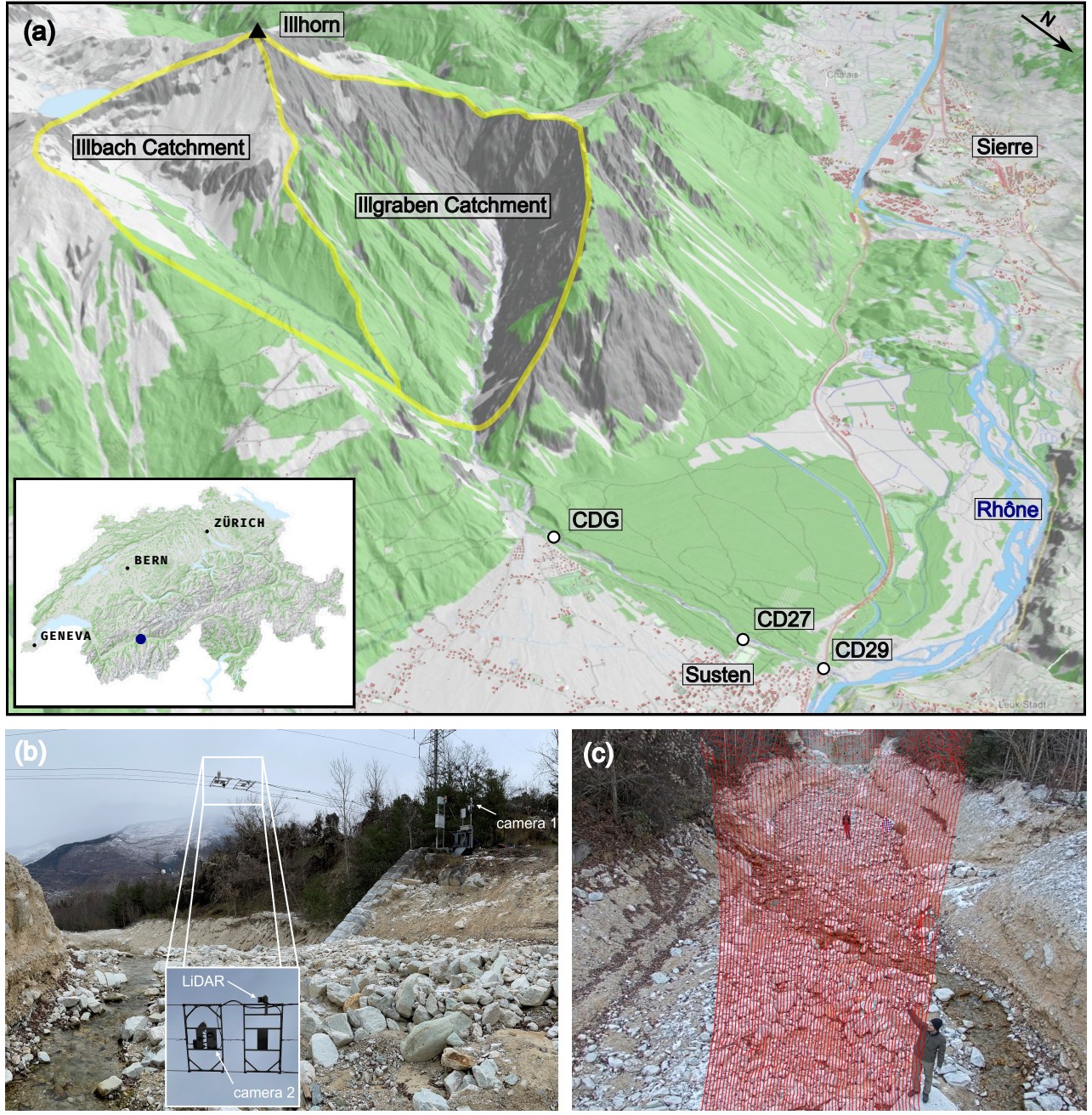

**Figure 1.** (a) Geographic overview of the Illgraben Channel (46.27° N, 7.61° E) in the Rhone Valley, Switzerland. The catchment boundary is outlined in yellow, and three monitoring sites (CDG, CD27, and CD29) are marked along the Illgraben channel, with inter-site distances of 1 270 m (CDG-CD27) and 470 m (CD27-CD29) (3D base map ©swisstopo). (b) Overview of the monitoring station (CDG) showing the placement of Camera 1, Camera 2 and a *Ouster* LiDAR Sensor system mounted on a suspended cableway structure (6-10 m above the check dam), facing downstream. (c) Visualization of the LiDAR scan coverage (red points) over the debris-flow channel, facing upstream.

2013; Berger et al., 2011a, b; Hürlimann et al., 2003; McArdell et al., 2007; McArdell, 2016). Recently, LiDAR scanners were installed 6-10 m above three of the check dams (CDG, CD27, and CD29; see Fig. 1b). The multi-beam LiDAR scanners in place during the event analyzed herein are *OS0* and *OS1* models from *Ouster*, operating at a sampling frequency of 10 Hz (see also Spielmann and Aaron, 2024; Spielmann et al., 2025; Aaron et al., 2025). These sensors record 3D point clouds of moving debris flows, which can then be further analyzed (e.g., Aaron et al., 2023; Zhu et al., 2023; Åberg et al., 2024; Hirschberg et al., 2025).

In this study, we examine a debris-flow event that occurred on 5 June 2022. The debris-flow event was characterized by a steep bouldery front (cf. Fig. 3) that arrived at CDG with a velocity of 5.5 m/s, slowing down to 3.4 m/s at CD27 and to 2.8 m/s at CD29, and the event lasted for around 30 minutes (see also Spielmann et al., 2024; Aaron et al., 2025). However, in this study, we only analyze the first 15 minutes of the event, which includes the majority of visible debris-flow features. The event further featured surge waves and rolling boulders throughout the event. Over 60 surge waves developed in the lower part of the fan such that they were only visible at CD27 and CD29, but not at CDG. Rolling boulders with the size approximately corresponding to the flow depth occurred throughout the event (Hirschberg et al., 2025).

## 3   Methods

Our study integrates 3D point-cloud data from LiDAR sensors and applies convolutional neural network (CNN) models to accurately detect debris-flow features at three monitoring stations along the Illgraben channel, demonstrating the potential of deep learning for hazard monitoring. Figure 2 illustrates a processing workflow to detect and track those objects with minimal labeling effort. The results derived from this workflow are compared with velocities obtained through a LiDAR-camera fusion method described by Hirschberg et al. (2025), showing that an alternative approach yields comparable results. Additionally, we incorporate velocity data from the same event, calculated using the Particle Image Velocimetry (PIV) method outlined by Aaron et al. (2023) and Spielmann et al. (2024). The PIV method generates dense surface velocity vector fields, which can then be averaged over a specific channel cross-section or rectangular area over time, providing a consistent surface velocity measurement for the selected section (Thielicke and Sonntag, 2021).

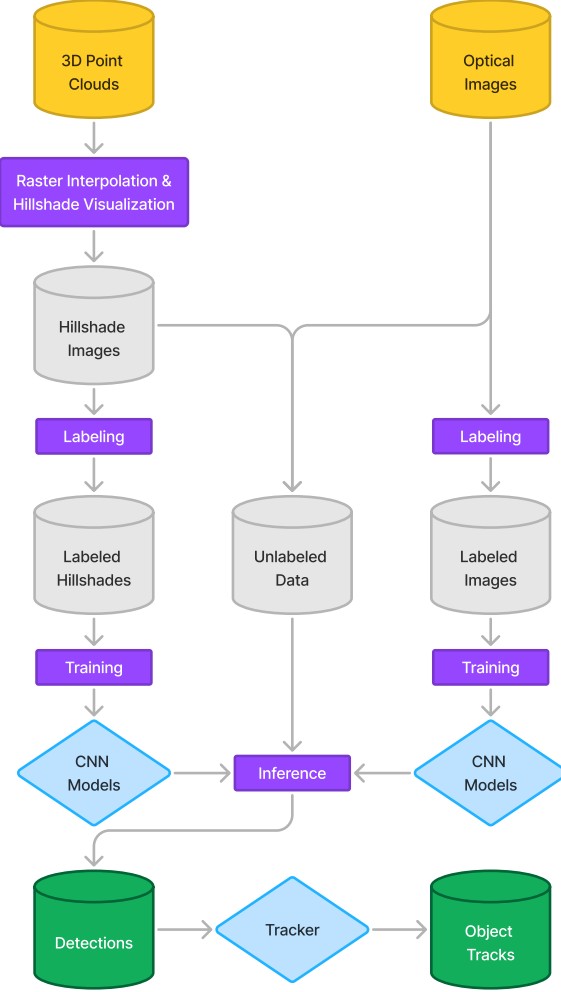

**Figure 2.** Workflow of the proposed method utilizing hillshade projections (left) and a LiDAR-camera fusion (right) as presented by Hirschberg et al. (2025). The process begins with the interpolation, rasterization, and hillshading of 3D point clouds collected from LiDAR sensors. These hillshade images are manually annotated by a human operator. Subsequently, six object detection models are trained using the labeled data. The trained models are applied to unlabeled data during inference, generating object detections and tracks. In the diagram, cylinders represent data inputs, while rectangles denote processing steps.

## 3.1 Hillshade image generation

To generate the hillshade images, the raw point clouds were first subjected to pre-processing. The coordinates of the point clouds were transformed into a coordinate system aligned with the channel ($x$-axis parallel to the channel, $y$-axis perpendicular, and $z$-axis vertical) as described in Spielmann and Aaron (2024). Corrections were applied to account for sensor wobbling caused by wind impacts on the LiDAR installation by performing a rigid iterative closest point (ICP) transformation that aligned successive scans through minimization of point-to-point distances in stable areas outside the channel bed. This correction was applied only to station CDG. Artifacts in the point clouds such as reflections off raindrops were removed by selecting a specific area of the point cloud that only corresponds to the section of the channel upstream of the check dam (Fig. 3b), where the resolution is highest and the results remain unaffected by the presence of the check dam. Additionally, to ensure consistent image dimensions, all point clouds were cropped using the same $x$ and $y$ limits. For creating the hillshade projections, the point clouds in .PLY format were imported to Python using the Open3D library (Zhou et al., 2018). A mesh was then generated with the aforementioned $x$ and $y$ limits and a grid spacing of 0.02 m to achieve a consistent 2D hillshade projection. Based on this mesh, the point-cloud data were interpolated and subsequently rasterized (Fig. 3c). The resulting raster model was visualized using a hillshading algorithm (Fig. 3d), implemented via the Relief Visualization Toolbox (Kokalj and Somrak, 2019; Kokalj et al., 2022). The azimuth angle was set to 90° and the altitude angle to 45°, as this configuration led to the highest contrast of the debris-flow topography. The final output consisted of hillshade images with a 2 cm pixel resolution, captured at frequency of 10 Hz.

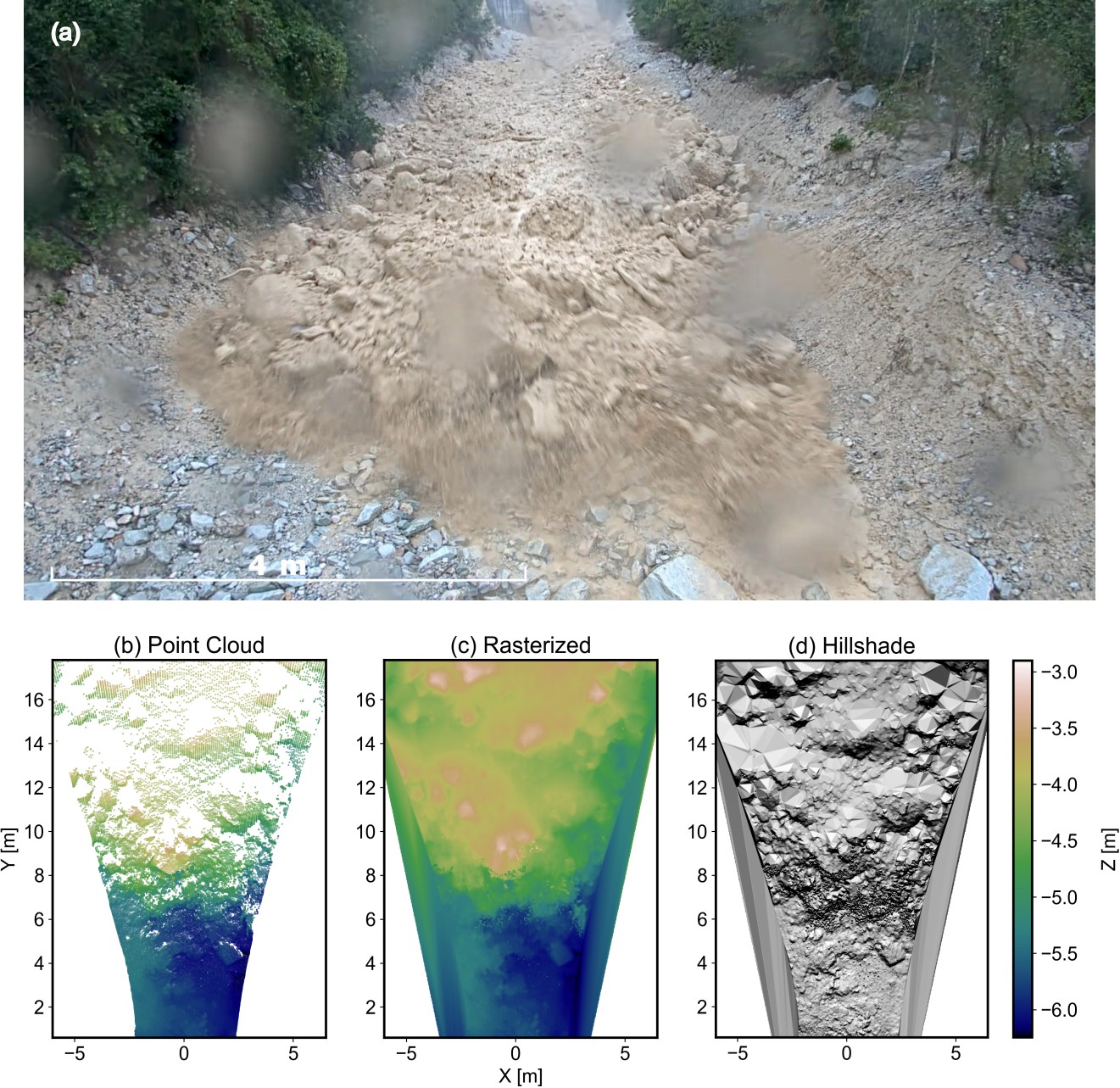

**Figure 3.** Multi-representation of debris-flow data. (a) Image capture (Camera 2) of an active debris-flow event (5 June 2022) in the Illgraben Channel, showcasing the flow front. (b) The corresponding 3D point cloud representation generated by the *Ouster OS1* LiDAR sensor illustrates the spatial distribution of points. (c) Rasterized and interpolated elevation model derived from the point cloud, providing a gridded representation of the debris-flow surface. (d) Hillshade visualization, emphasizing terrain features through simulated shading for enhanced interpretation of morphological characteristics.

### 3.2 Labeled dataset of hillshade projections

To ensure a consistently labeled dataset, all annotations were created by a single individual under the supervision of experts with substantial knowledge of the domain and study site. The open-source data labeling platform, Label Studio (HumanSignal, 2024), was used for this task. Additional video imagery was used as a reference during the labeling process to accurately identify debris-flow objects. When objects were considered significant enough to be classified into one of the four categories (Fig. 4), a rectangular box was drawn around the object of interest, referred to as a label hereafter. Significance was defined as the ability to distinguish the object from the background noise in the hillshade image. The objects of interest were defined and tested in previous studies by Aaron et al. (2023); Spielmann and Aaron (2024); Hirschberg et al. (2025), and these definitions were adopted in this work. The four feature classes include Boulders, Surge Waves, Rolling Boulders, and Woody Debris (Fig. 4). These object types are hypothesized to represent distinct vertical velocity profiles within the debris flow. Woody debris, which floats on the surface, is assumed to travel at the surface velocity. In contrast, rolling boulders, which are approximately of the same size as the flow depth and remain in contact with the channel bed, rather represent the depth-averaged velocity (Aaron et al., 2023). To further investigate these properties, we retained the same feature classes. Hillshade images for model training were randomly selected from all three monitoring stations and two events (19 September 2021, 5 June 2022), with additional images included for more complex scenarios, such as the debris-flow front. A total of 1 092 images were selected, with 21 698 instances annotated (Tab. 1). The entire dataset was then divided into training, validation, and testing sets with a ratio of 70:15:15.

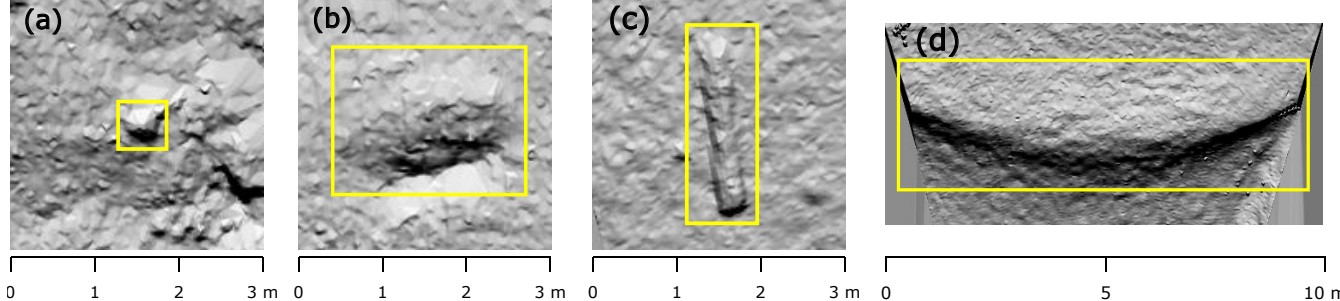

**Figure 4.** Example annotations of the four feature classes used in the object detection training set. (a) Boulder: small to medium-sized sliding rocks. (b) Rolling Boulder: a displaced rock structure indicating rolling movement, with characteristic turbulence around the object. (c) Wood: an elongated object resembling a log, occasionally including branches or roots. (d) Surge Wave: a surge traveling through the channel over the entire channel width.

**Table 1.** Summary of our dataset, detailing the number of images and annotated instances across the training, validation, and test sets for all classes combined (Boulder, Surge Wave, Rolling Boulder and Wood). The dataset was split with a ratio of 70:15:15.

| Set | Class | # Images | # Instances |
|-----|-------|----------|-------------|
| Training | all | 764 | 14 914 |
| Validation | all | 164 | 3 308 |
| Test | all | 164 | 3 476 |

## 3.3 Object detection and tracking

We tested and compared four state-of-the-art object detection models, including three versions of YOLO (v5, v8, and v11), as well as RT-DETR, RetinaNet, and Faster R-CNN, using our custom hillshade dataset. The models were selected based on their popularity and performance, all ranking among the top performers in the COCO benchmark (cf. Zou et al., 2023). In addition, we evaluated multiple architectures to assess their ability to accurately detect small and partially occluded objects, as model performance can vary substantially across object scales (Zou et al., 2023). The largest pre-trained versions of each YOLO model (YOLOv5-X, YOLOv8-X, YOLOv11-X) and RT-DETR-X were deployed, along with a pre-trained ResNet-50 backbone for RetinaNet50 and a ResNeXt-101 backbone for Faster-RCNNX101. The evaluation was conducted using the same dataset, with identical training, validation, and test splits across all models. Training and detection processes were conducted on NVIDIA GeForce RTX 4090 GPUs available through the Euler IX cluster at ETH Zürich.

### 3.3.1 Object detection models

Detectron2 is a highly modular framework developed by Meta AI Research for state-of-the-art object detection and segmentation tasks. Built on PyTorch, it provides powerful tools for a wide range of applications and features an extensive model zoo with pre-trained weights. Using this framework, we evaluated variations of the RetinaNet (ResNet-50 and ResNet-101) and Faster R-CNN models (ResNet-50, ResNet-101 and ResNeXt-101), where 50 and 101 denote the depth of the backbone network (Wu et al., 2019).

Faster R-CNN is a widely adopted two-stage object detection model that builds upon the success of its predecessors, R-CNN and Fast R-CNN. The key innovation of Faster R-CNN is the introduction of the Region Proposal Network (RPN), which efficiently generates region proposals as part of the model, significantly speeding up the detection process. In the first stage, the RPN proposes candidate object regions, and in the second stage, these proposals are refined and classified into object categories. The model's ResNet backbone is responsible for feature extraction from the input image, while the RPN and the detection head handle proposal generation and final classification, respectively (Ren et al., 2016).

RetinaNet is a widely used one-stage object detector, developed when two-stage detectors within the region-based CNN framework were dominant. The key difference between one-stage and two-stage detectors is that one-stage detectors predict objects and their bounding boxes directly from the image in a single pass, whereas two-stage detectors first generate region-based proposals and then refine them in a second stage. RetinaNet addresses the problem of class imbalance by introducing

Focal Loss, a function that reduces the weight of well-classified examples, allowing the model to focus on harder-to-classify, minority class examples without sacrificing performance on the majority class. The model also consists of a ResNet-based backbone (FPN) and two task-specific subnetworks: one for object classification and the other for bounding box regression (Lin et al., 2018).

Building on the advancements in one-stage detectors like RetinaNet, YOLO (You Only Look Once), originally introduced by Redmon et al. (2016), optimized detection speed and accuracy through its unified detection architecture. YOLOv5 (Jocher, 2020) further improved upon this by introducing multi-scale detection, data augmentation, and AutoAnchor algorithm, which search for ill-fitted anchor boxes. It features the CSPDarknet53 backbone and FPN for feature extraction, balancing accuracy and speed effectively (Terven et al., 2023). YOLOv8 (Jocher et al., 2023) builds on YOLOv5 by introducing the C2f module for improved detection accuracy through better feature-context integration. It adopts an anchor-free design with a decoupled head for objectness, classification, and regression tasks, enhancing precision (Terven et al., 2023). The latest YOLO version, YOLOv11, incorporates new architectural elements like the C3k2 and partial Pyramid Pooling - Fast (SPPF) block. The C3k2 module improves feature representation by deepening the network's capacity to capture intricate patterns. Meanwhile, the SPPF block accelerates the model by pooling features at multiple scales, enabling better handling of objects of varying sizes with minimal computational overhead Jocher et al. (2024).

Most recent object detectors, such as the YOLO series, use a Non-Maximum Suppression (NMS) algorithm to filter overlapping predictions, such as bounding boxes in object detection. NMS removes predictions below a probability threshold and selects the highest-probability entity, discarding others with an Intersection over Union (IoU) above a specific threshold (typically 0.5). However, NMS slows down inference and introduces hyperparameters that destabilize both speed and accuracy (Zhao et al., 2024). Recently, a new object detection model was introduced by Zhao et al. (2024), presenting a Transformer-based detector. The authors suggest that their Real-Time Detection Transformer (RT-DETR) can address these issues. Specifically, by separating intra-scale interactions (features within the same scale) and cross-scale fusion (features from multiple scales) to enhance speed, an effective hybrid encoder is developed that efficiently processes multi-scale data. The uncertainty-minimal query selection, which minimizes the discrepancy between classification and localization confidence, is proposed to provide the decoder with high-quality initial queries, thereby improving accuracy (Zhao et al., 2024).

For the evaluation of the trained models, we employed the widely used object detection metrics: Precision (P), Recall (R), F1-Score, and mean Average Precision (mAP). P represents the accuracy of positive predictions, while R reflects the model's sensitivity. The F1-Score is the harmonic mean of P and R. The mAP is calculated by integrating the P-R curve across various Intersection over Union (IoU) thresholds (commonly 0.5 and 0.5 to 0.95). A more detailed definition of these metrics can be found in the Appendix A1.

### 3.3.2 Data augmentation

The choice of data augmentation techniques can significantly impact model performance and robustness (Shorten and Khoshgoftaar, 2019). To address class imbalance, we implemented a basic data augmentation strategy by horizontally flipping images of underrepresented classes. The feature class boulder accounts for the majority of instances in the training dataset, with 13 882

occurrences, while surge wave, rolling boulder, and wood are represented by only 73, 459, and 500 instances, respectively. To mitigate this imbalance, we selected images containing one or more of these underrepresented classes and reintegrated them into the training dataset. This augmentation approach was consistently applied across all tested models. Certain models, particularly those optimized for fine-tuning on small amounts of task-specific data, rely on extensive data augmentation to enhance their performance (e.g., YOLOv5). In contrast, most popular implementations of Transformer-based backbones, such as those in Detectron2, provide a default set of augmentations designed for large-scale pretraining, typically limited to random flips, resizing, and cropping (Ruis et al., 2024). For this study, we applied augmentations which were meaningful to our application, including translation, mosaic, and resizing, while omitting augmentations like RGB saturation due to the use of grayscale images. These choices align with the default augmentations recommended by the respective model developers. Specifically, we employed techniques such as Mosaic and Mixup for YOLO models, and image resizing for Detectron2-based models (Tab. 2).

**Table 2.** Summary of image augmentation techniques and their corresponding parameters applied across all models, including RT-DETR, YOLO versions, and the Detectron2 models Faster-RCNN and RetinaNet. Horizontal flipping was applied exclusively to images containing underrepresented feature classes (Surge Wave, Rolling Boulder, and Wood).

| Model | Augmentation | Value/Probability |
|---|---|---|
| All models: | Horizontal Flipping | 0.5 |
| YOLOv5, YOLOv8, YOLOv11, RT-DETR: | Translate | 0.1 |
| | Scale | 0.5 |
| | Mosaic | 1 |
| | Mixup | 0.5 |
| Faster-RCNN, RetinaNet: | Resize Shortest Edge | (640, 672, 704, 736, 768, 800) |
| | Scale Jittering | 0.5 |

### 3.3.3 Multi-object tracking

After detecting debris-flow features on individual hillshade images, we employed two Multi-Object-Tracking (MOT) algorithms to unify the detections across subsequent frames. The Simple Online and Realtime Tracking (SORT) algorithm (Bewley et al., 2016; Bewley, 2024) predicts the movement of detected objects between frames using a linear constant velocity model and refines these predictions with a Kalman Filter. To associate predicted positions with actual detections, SORT utilizes the Hungarian algorithm, optimizing the assignment based on an IoU metric. Each object is assigned a unique identifier, which is maintained across frames if the association is successfully established. However, preliminary testing showed that the SORT algorithm performed poorly for objects with velocities greater than 5 m/s. To overcome this limitation, we applied the BoT-SORT algorithm (Aharon et al., 2022). BoT-SORT builds on the tracking-by-detection approach used in SORT but incorporates an improved Kalman filter state vector for more accurate box localization and a camera motion compensation feature.

### 3.4 Object velocity and grain size

The resulting tracks from the MOT algorithms were used to calculate object velocities. First, we converted the bounding box coordinates from pixel units to meters using the hillshade image resolution of 0.02 m/px. The (instantaneous) velocity was then calculated based on the temporal displacement of the bounding box centroid between consecutive frames (Eq. 1).

$$v(t_i) = \frac{\sqrt{(x_c(t_i) - x_c(t_{i-1}))^2 + (y_c(t_i) - y_c(t_{i-1}))^2}}{t_i - t_{i-1}} \tag{1}$$

where $v_{t_i}$ is the velocity of a tracked debris-flow feature and $x_c$ and $y_c$ are the centroids object coordinates at time $t$. The time step between two frames is 0.1 s, corresponding to the LiDAR recording frequency of 10 Hz.

This approach is less sensitive to uncertainties than using bounding-box edges, which can be influenced by object shadows. For object tracks that experienced a class change, we assumed that the most frequently occurring class corresponded to the ground truth. In addition, we also calculated the average-track velocities by averaging all instantaneous velocities of a track (object).

In addition to object velocity, a grain size analysis was conducted using the width of the detected bounding boxes as a proxy for the characteristic grain size of individual objects. In the following, the term grain size refers to the bounding box width. The limitations of this simplified assumption will be discussed below (Sect. 5).

## 4 Results

### 4.1 Metrics and model assessment

Table 3 presents a comparison of P, R, F1-score, and mAP at IoU thresholds 0.5 (mAP50) and 0.5-0.95 (mAP50-95) across the four feature classes (cf. Fig. 4) on the test dataset. Overall, YOLOv8-X stands out in precision, while YOLOv11-X tops in mAP50-95, indicating best performance across varying IoU thresholds. Notably, differences in metrics are minor, with variations of <0.067 in mAP and <0.054 in F1-score across all models. However, the class-averaged metrics are disproportionately influenced by the high precision values of the surge wave class (e.g., P=1 and R=0.988 with the YOLOv5-X model). These trends are also evident when analyzing the metrics by class (see Fig. A1, Tab. A1). The model comparison produced results consistent with the previous observations, with RetinaNet50 showing the lowest performance (particularly for the Boulder class, with mAP50-95=0.382) and YOLOv11-X demonstrating the best performance, showing strong mAP50-95 values across all classes. When analyzing the performance metrics across varying confidence thresholds (CT), it is observed that YOLOv5-X, YOLOv8-X, and YOLOv11-X achieve high F1 scores (>0.8) at a low CT value of 0.1 but exhibit a sharp decline when the CT exceeds 0.7 (Fig. 5). In contrast, RT-DETR-X, Faster-RCNN, and RetinaNet reach their peak performance at higher CT values (0.6-0.7) but demonstrate poor precision and recall at lower CT thresholds. Inference speed was tested on the hillshade projections for the CDG station at a CT of 0.1 and an IoU threshold of 0.5. With a processing speed of 38.2 frames per second

(FPS), YOLOv11-X processes nearly at double speed compared to the other models which all have similar FPS between 16.1 and 21.7 (Tab. 3).

**Table 3.** Performance metrics for object detection of debris-flow features (Boulder, Surge Wave, Rolling Boulder, and Wood) on the test dataset, ordered by mAP50-95. Metrics include Precision (P), Recall (R), F1 score, mean average precision at an intersection over union threshold of 0.5 (mAP50), and across thresholds from 0.5 to 0.95 (mAP50-90). A confidence threshold of 0.1 and an IoU threshold of 0.5 were applied. Class-specific metrics are listed in the Appendix Tab. A1. FPS (Frames Per Second) describes the number of frames a model can process in one second, indicating its inference speed and real-time capability, including pre- and post-processing (i.e., loading and saving data).

| Model | P | R | F1 | mAP50 | mAP50-95 | FPS |
|---|---|---|---|---|---|---|
| YOLOv5-X | 0.866 | 0.792 | 0.827 | 0.859 | 0.579 | 17.1 |
| YOLOv8-X | 0.890 | 0.769 | 0.825 | 0.859 | 0.576 | 21.7 |
| YOLOv11-X | 0.817 | 0.823 | 0.820 | 0.862 | 0.583 | 38.2 |
| RT-DETR-X | 0.833 | 0.799 | 0.816 | 0.851 | 0.545 | 16.9 |
| Faster-RCNNX101 | 0.848 | 0.796 | 0.821 | 0.828 | 0.519 | 17.7 |
| RetinaNet50 | 0.794 | 0.753 | 0.773 | 0.795 | 0.506 | 16.1 |

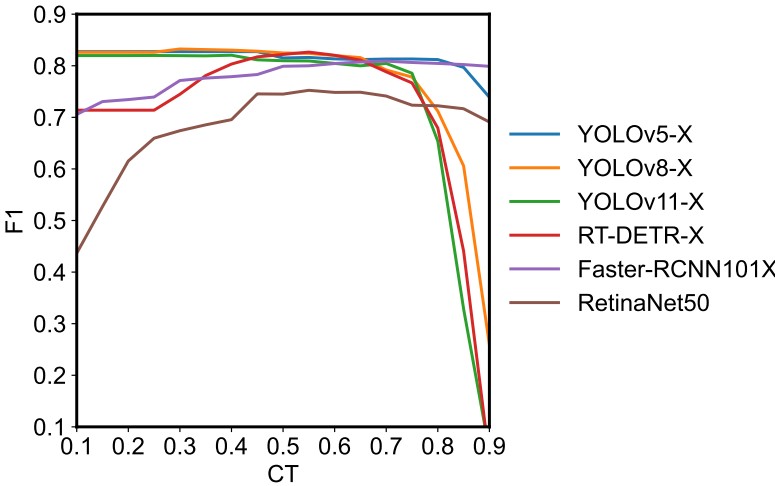

**Figure 5.** Evaluation of F1 Score sensitivity to Confidence Threshold (CT) for YOLOv5-X, YOLOv8-X, YOLOv11-X, RT-DETR-X, Faster-RCNN101X and RetinaNet50

## 4.2 Object track analysis

### 4.2.1 Object track velocities

The computation of object velocity using frame-by-frame tracking with BoT-SORT, provides a detailed dataset of object velocities for each timestep and detected object. The velocities were analyzed either as instantaneous or track-average velocities. Object tracks with an average velocity below 1 m/s were excluded, as their low velocity indicates (temporary) deposition. A total of 23 120 object tracks were recorded and reduced to 16 452 after applying the described filtering (Tab. 4). For the three stations, between 25% and 32% of object tracks were filtered and the number of object tracks decreases from upstream to downstream by 70%. For a comparison, we also show the number of detected objects on camera images (LiDAR-camera fusion: Hirschberg et al., 2025). Interestingly, the number of detections shows a different trend among the stations.

**Table 4.** Summary of detections and object counts for all stations (CDG, CD27, and CD29). It includes the total number of detections, the number of unique objects identified on hillshade projections, the number of filtered hillshade objects (after post-processing), and the number of objects identified from the LiDAR-camera fusion method (Objects camera: Hirschberg et al., 2025).

| Station | Class | # Detections | # Objects LiDAR | # Objects LiDAR (filtered) | # Objects Camera |
|---------|-------|--------------|-----------------|----------------------------|------------------|
| CDG | all | 262 405 | 12 037 | 8 191 | 523 |
| CD27 | all | 142 298 | 7 959 | 5 882 | 1 005 |
| CD29 | all | 98 713 | 3 124 | 2 379 | 990 |
| Total | all | 503 416 | 23 120 | 16 452 | 2 518 |

In Figure 6, these object (track-average) velocities are compared to the results from the PIV hillshade surface velocity measurement method. At CDG, the velocities of objects detected by LiDAR were consistently slightly higher compared to the PIV velocities, whereas at CD27 and CD29, both velocity measurements showed good agreement. Additionally, at CD27 and CD29, the LiDAR velocities exhibited little variation and closely followed the PIV velocities. In contrast, at CDG, the LiDAR velocities displayed significant variability, with differences of up to 2 m/s within the same timestep. Overall, the velocities showed a gradual downstream decrease from CDG to CD29. As shown in Figure 6, boulders were predominantly detected at the beginning of the event, while rolling boulders peaked later and persisted towards the end, especially for CDG and CD27. Woody debris, although less frequently detected, appeared consistently throughout the event at all stations. The agreement between object velocities obtained using the LiDAR method and the LiDAR-camera fusion is illustrated in Fig. A2.

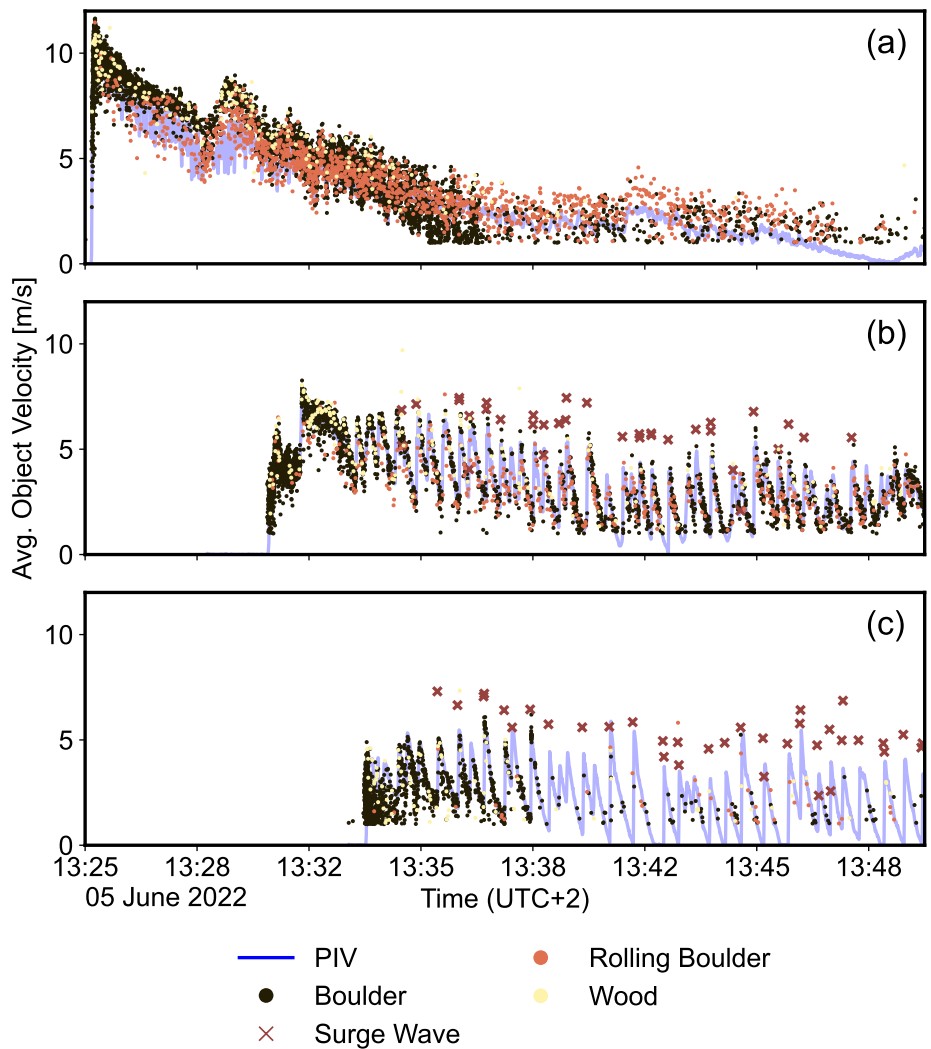

**Figure 6.** Mean object velocities calculated for the 5 June 2022 event, using the LiDAR hillshade method at the three monitoring stations: (a) CDG, (b) CD27, and (c) CD29. Colors represent the different object classes. Surface velocities derived from particle image velocimetry (PIV) applied to LiDAR hillshades by Spielmann et al. (2024) are included as a reference.

## 4.2.2 Grain sizes and flow depth

Grain size comparisons relative to flow depth at each station highlighted notable differences, as indicated in Fig. 7. At CDG, a substantial proportion of tracked objects was larger than the flow depth, while at CD29, nearly all tracked objects were smaller than the flow depth (Fig. 7). Furthermore, the absolute grain size shows a decreasing trend from CDG to CD29, with many objects exceeding 2 m at CDG and almost all objects being smaller than 2 m at CD29. Notably, the flow depth increased from CDG to CD29, indicating that some objects might be submerged and not visible at CD29, which will also be discussed below (Sect. 5). The detection frequency is highest with the front arrival 1-2 subsequent minutes (Fig. 7) and continuously decreases thereafter, except for CDG, where more objects are detected again when flow depth decreases to $\sim 1$ m.

We validated the detected grain sizes by comparing them to the size of manually measured object widths in individual frames (Fig. A4 Appendix). The offset of 10-20% can be explained by the labeling procedure, in which training boxes were intentionally drawn larger to include surrounding water turbulence or shadows which help to identify objects (cf. Figure 4b).

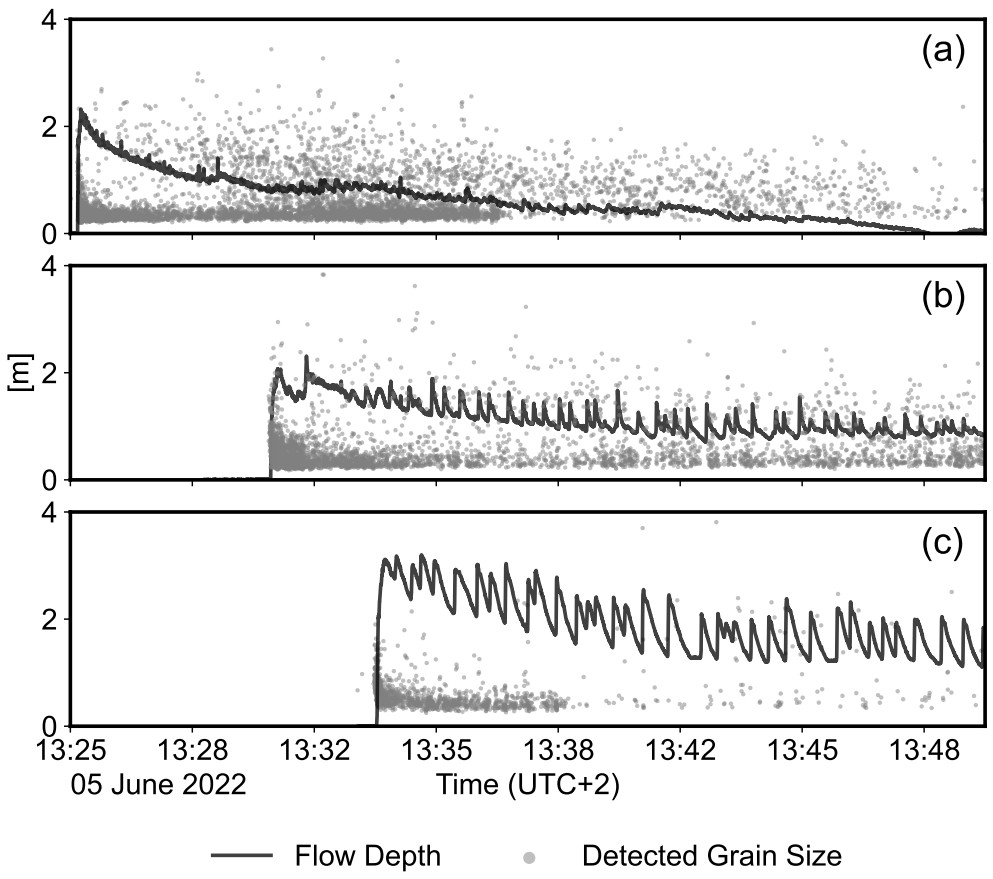

**Figure 7.** Comparison of flow depth and grain size (boulders and rolling boulders) over time at the three monitoring stations (a) CDG, (b) CD27, and (c) CD29 during the 5 June 2022 event. The black line represents the flow depth (in meters), while the gray markers show LiDAR-detected grain sizes, averaged across individual object tracks.

### 4.3 Grain size distribution and longitudinal sorting

As indicated in Figure 7, the observed boulder and rolling boulder sizes vary over time and location. This allows for studying dynamic grain size distributions (GSDs) along the flow path, as summarized in Fig. 8, where LiDAR detections (Fig. 8a-c) represent all target sizes in a given frame (prior to object tracking), while LiDAR tracks (Fig. 8d-f) indicate the size of individual objects across multiple frames (after unification by the tracking algorithm).

At station CDG, the GSD becomes coarser over time (Fig. 8a/d), whereas at station CD29, the trend is reversed, with finer grain sizes observed as time progresses (Fig. 8c/f). Station CD27, located between CDG and CD29, exhibits a less pronounced trend of coarsening over time (Fig. 8b/e). The trends of coarsening at CDG, fining at CD29, and a combination of fining and coarsening at CD27 are evident in both the detection and tracking data. The particle sizes at stations CDG and CD27 (see Tab.

A2) are comparable across both the detection and object track datasets, with $D_{50}$ (median grain size) values ranging from 0.36 to 0.40 m at CDG and from 0.36 to 0.39 m at CD27. In contrast, particle sizes at station CD29 are generally larger than those at CDG and CD27, with $D_{50}$ values ranging from 0.46 to 0.50 m. At stations CD27 and CD29, $D_{50}$ becomes progressively finer over time. In contrast, $D_{50}$ at station CDG shows minimal change. Notably, at CDG, the $D_{90}$ ($90^{th}$ percentile) becomes coarser over time, likely due to the increased presence of rolling boulders after 13:30 UTC+2 (cf. Fig. 6), a process less prominent at

CD27 and CD29.

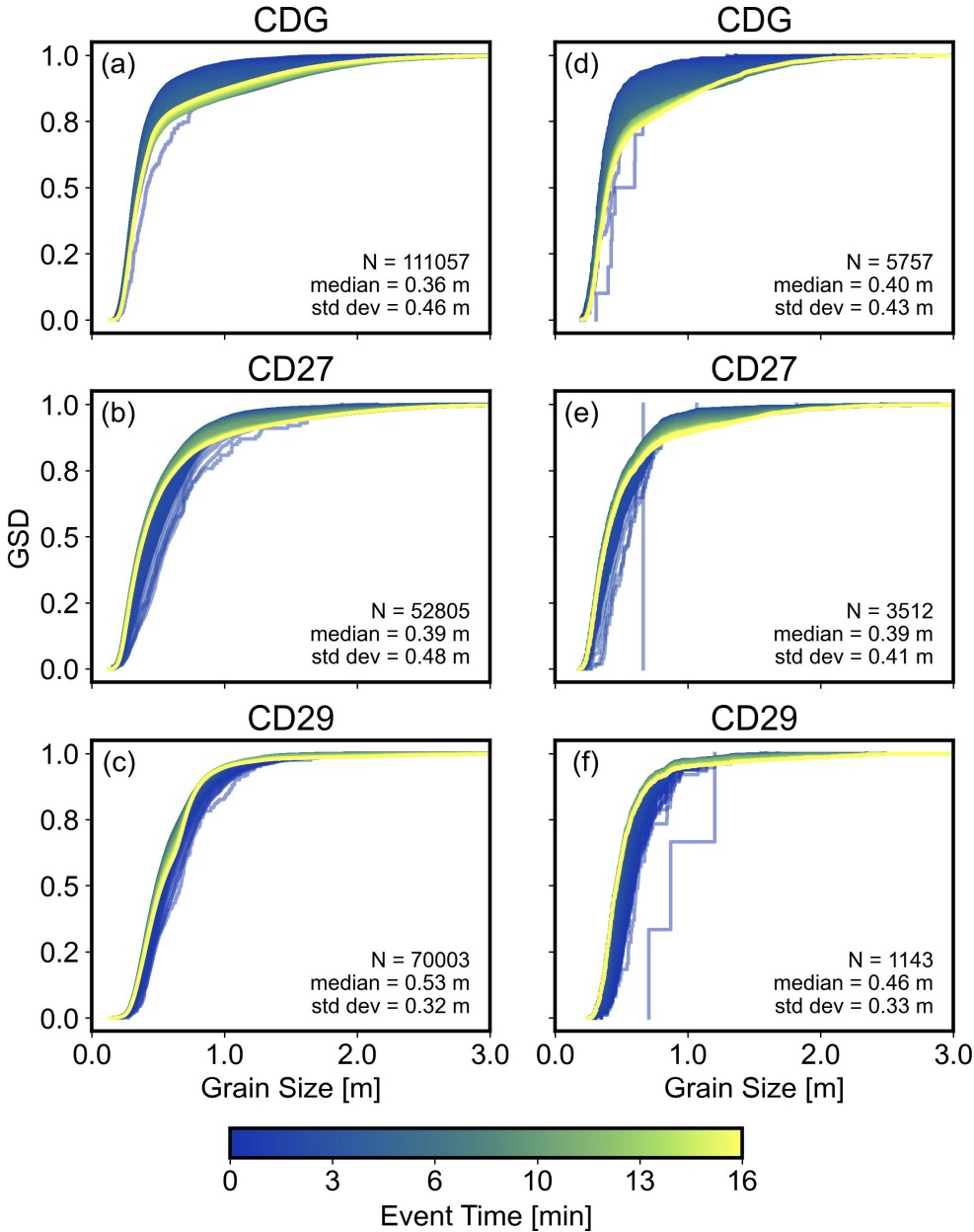

**Figure 8.** Grain size distributions (GSD), shown as empirical cumulative distribution functions, across stations CDG, CD27, and CD29. GSDs derived from frame-by-frame detections (a-c) and object tracks (d-f). GSDs are colored by event time to show temporal evolution, and each line represents the GSD of all detected and tracked objects until the current event time, respectively. Event start time is defined by the arrival of the debris-flow front at the check dam.

# 5 Discussion

## 5.1 Performance of object detection models

We present a novel automated method to extract relevant information on debris-flow features from high-resolution 3D LiDAR measurements at three monitoring stations along the Illgraben channel. By converting raw LiDAR point clouds into hillshade projections and applying object detection algorithms to these 2D images, our approach enables the accurate identification of debris-flow objects. For the debris flow analyzed in this study, about 16 000 objects were automatically identified, which would take a human operator approximately 3 555 hours to extract (assuming 16 000 objects, 40 frames per object, 20 s annotation time). The detected objects can then be linked to reconstruct tracks of boulders, woody debris, and surge waves, allowing for the calculation of their velocities. This framework offers a foundation for efficient data processing at other locations and facilitates the analysis of various debris-flow events across diverse settings. Furthermore, the framework is transferable independent of ambient light conditions. Unlike standard cameras, it is therefore applicable both day and night. Additionally, it is significantly more efficient than working directly with 3D data and can potentially be operated in real time. Therefore, we present a basis for developing novel early-warning systems.

We tested six different object detectors on their ability to detect debris-flow features. The performance metrics of the different models showed very similar results (Tab. 3). Therefore, all tested model architectures are equally suited for debris-flow object detection. The small variations in model accuracy suggest that the limiting factor is the quality and size of the dataset rather than the detector architecture or hyperparameter optimization (Hirschberg et al., 2025). The inference speed, however, reveals major differences between models up to a factor of $\sim 2.5$. Important considerations when selecting a model are therefore the computational and speed requirements, especially for real-time applications. Single-shot detectors, especially YOLOv11-X, confirmed to be desirable in this regard (Jocher et al., 2024).

The resulting precision and recall metrics, both approximately 0.8, are generally consistent with results reported in other applications, where values between 0.7 and 0.9 are considered good (Diwan et al., 2023). The slightly lower performance may be attributed to the large number of small objects detected, which are inherently more challenging to identify. This difficulty arises because CNN architectures downsample feature maps across several levels, resulting in a loss of spatial information at smaller scales (Li et al., 2020; Liu et al., 2021). Consequently, the precision values for the boulder class, the smallest among the four classes, were notably lower (Fig. A1, Tab. A1).

Because the split into training and testing datasets was done randomly, images containing the same objects may end up in both datasets. Therefore, some information leakage from the training to the testing dataset is likely. As a consequence, the recall metrics may be inflated. However, this inflation should be negligible because such an effect was excluded through systematic testing by splitting by frame number rather than randomly for debris-flow object detection on optical images (Hirschberg et al., 2025). In summary, the metrics are expected to be representative and demonstrate good performance for all classes.

## 5.2 Implications for debris-flow mechanics

Although longitudinal sorting is argued to control key hazard parameters such as peak debris-flow discharge (Hungr, 2000), the mechanism is difficult to analyze in the field. We aimed to investigate how GSDs evolve throughout a debris-flow event and along a fan. The $D_{50}$ at CD27 and CD29 showed fining as expected, but slight coarsening at CDG (Fig. 8). $D_{90}$ coarsened at all stations, but the degree of coarsening decreased from upstream to downstream. This could indicate that the longitudinal profile of coarser particles is not yet mature in the upper parts of the fan and that sorting is still ongoing. Several aspects of our measurements support this interpretation: a) there are many detections later in the event at the upstream station CDG (Fig. 7); b) front velocities decrease from upstream to downstream (Fig. 6); and c) many objects travel downstream faster than the front, in particular at CDG (Fig. A3). Overall, this suggests that the many boulders detected later in the event at CDG could catch up with the front as they travel downstream (see also "centerline advection" Spielmann et al., 2025). The accumulation of boulders at the front could also explain the increase in flow depth along the flow path (Fig. 7), which cannot be explained by changes in cross-section geometry, but is also influenced by changing slope angles (Aaron et al., 2025).

The number of detections decreases from upstream to downstream (Fig. 8). This could indicate that some of the larger boulders detected later in the event at CDG may become submerged downstream as flow depth increases. For CD27, the lower number of detections could also be due to technical reasons, as the LiDAR sensor had a lower spatial resolution. However, the detection of larger boulders should not be strongly affected. In addition, the deposition of many large boulders along the flow path is unlikely in a confined and reinforced channel. Overall, we believe that the decreasing front velocities and higher object velocities likely indicate an ongoing sorting process that may lead to increased flow depth, which in turn submerges objects detected upstream, contributing to a mass imbalance. The presented method will be applied to additional events to validate these findings.

Finally, the ability to retrieve GSD data from natural debris flows is critical for improving runout models (e.g. Piton et al., 2022). As highlighted by de Haas et al. (2015), runout and deposition are strongly influenced by flow composition, including water content and the presence of large debris. They further suggest that incorporating GSD into debris flow models could significantly enhance predictive capabilities – an aspect that our method could address effectively.

## 5.3 Limitations

The object velocities derived from feature tracking were compared with velocities computed for the same event using PIV by Aaron et al. (2025), showing strong overall agreement across all three monitoring stations (cf. Fig. 6). However, at station CDG, between 13:25 and 13:30 UTC+2, the estimated object velocities for the boulder class are approximately 1-2 m/s higher than the PIV velocities, whereas the rolling boulder class velocities closely match the PIV values. Higher peak velocities were also observed at CD27 and CD29 (Fig. 6b-c) compared to the PIV velocities, particularly for the surge wave class. This discrepancy is expected, as the reported PIV velocity is calculated as the mean within a 2 m × 2 m box of the velocity field, while object tracking directly follows the actual movement of individual features. The comparison of PIV velocity validates

the computation of object velocities based on the bounding box centroids. Furthermore, the computed velocities align closely
with object velocities (cf. Fig. A2) obtained through the LiDAR-camera fusion method by Hirschberg et al. (2025).

Another uncertainty in the computation of object velocities concerns the quality of multi-objects tracking (MOT), where validation is particularly challenging due to object occlusion and re-identification issues that lead to class switches or fragmented trajectories (Luo et al., 2021). The variability in performance metrics, lack of standardized ground truth, and reproducibility issues are well-known challenges in existing benchmark datasets (Milan et al., 2013). Furthermore, established MOT benchmarks such as Milan et al. (2016) focus on persons or vehicles rather than debris-flow objects. Therefore, as mentioned above, the presented object velocities were validated against the flow surface velocities inferred from PIV with satisfying results.

Stationary objects in the channel prior to the debris-flow front and objects temporarily deposited on the channel banks were filtered, with re-entrainment by subsequent surge waves accounted for in the analysis. It is important to note that when interpreting the velocity results, measured velocities are restricted to the LiDAR field of view, which partly does not cover the entire channel width (cf. Figure 1c). Nevertheless, the central 5 m of the channel are covered over a length of $\sim$20 m (Fig. 3). Another limitation arises from the conversion of 3D LiDAR point clouds into 2D hillshade images, which leads to the loss of spatial information along the vertical axis. As a result, the computed object velocities are constrained to a 2D plane, despite the fact that boulders may submerge and re-emerge, indicating vertical velocity components that remain uncaptured. However, the impact of this limitation is expected to be small, as the vertical velocity component is minor compared to the horizontal one. This is due to the low channel inclination on the fan (3–4$°$), and the 2D object velocities correspond well with the PIV-derived velocities (cf. Fig. 6/A2), which represent full 3D measurements (Aaron et al., 2025). 3D velocities could, however, be calculated by projecting the bounding box centroids on the 3D point clouds.

## 6 Conclusions

The new processing method presented in this study successfully implements automatic object detection for the efficient analysis of high-frequency LiDAR point clouds, enabling the calculation of thousands of debris-flow objects, such as boulders, rolling boulders, surge waves, and wood. By leveraging object detection models, the processing time is significantly reduced compared to manual analysis. Our results demonstrate that high-frequency point clouds (10 Hz), combined with a hillshading algorithm, can serve as input for traditional image-based object detection models, achieving high detection accuracies with only a fraction of the dataset labeled. Performance metrics across models show minimal variation. Notable differences are observed in processing times, with YOLOv11 achieving the highest frame rate at 38.2 FPS, and would therefore allow for real-time operation. The use of four distinct object classes proved effective and sufficiently distinct for accurate detection and provided the opportunity to gain deeper insights into flow dynamics. The computed velocities align well with values obtained from alternative methods for the same event, validating the approach. Additionally, the exclusive use of LiDAR data captures a greater number of objects and enables longer tracking durations over more frames, with the added advantage of functioning effectively during no- or low-light conditions, unlike standard cameras. During an ongoing event, a fining process is observed at all stations for smaller grains ($D_{50}$) and a coarsening for larger grains ($D_{90}$). The coarsening is more pronounced upstream

than downstream and object velocities exceed front velocities. On one hand this may indicate longitudinal sorting, on the other hand larger boulders may be submerged as flow depth increases. Future research will apply this method to more events to validate and strengthen the findings. Finally, the method holds promise for early-warning system developers due to its suitability for real-time implementation.

*Data availability.* Hillshade images are available under the ETH Research Collection at https://www.research-collection.ethz.ch/entities/researchdata/d5146581-0a73-4a12-bef3-0c1f10ac2f42 under a Creative Commons Attribution-ShareAlike 4.0 International license. The raw data (3D point clouds) are available in the data repository by Aaron et al. (2025) under https://doi.org/10.3929/ethz-b-000736836.

*Video supplement.* Video material of the event is available in the Supplementary Information to Aaron et al. (2025) under https://doi.org/10.3929/ethz-b-000736836.

## Appendix A

### A1 Evaluation of object detection

Generally, the performance of object detectors is measured by Precision (P) and Recall (R). P describes the accuracy of positive predictions (Equation A1) while R expresses the sensitivity of the model (Equation A2).

$$P = \frac{TP}{TP + FP} \tag{A1}$$

$$R = \frac{TP}{TP + FN} \tag{A2}$$

where TP refers to True Positives, FP to False Positives and FN to False Negatives. To asses both P and R, the F1 score is commonly used. It represents the harmonic mean of P and R, providing a balanced single metric (Equation A3).

$$F1 = \frac{2 \cdot TP}{2 \cdot TP + FP + FN} = \frac{2 \cdot P \cdot R}{P + R} \tag{A3}$$

A prediction of the object detection model is considered a TP, if the Intersection over Union (IoU) metric exceeds a defined threshold, 0.5 in our case. The IoU measures the overlap between the predicted and ground truth (GT) bounding box, shown in Equation A4.

$$IoU = \frac{Box_{GT} \cap Box_{Pred}}{Box_{GT} \cup Box_{Pred}} \tag{A4}$$

To evaluate a classifiers performance over all confidence scores, the Average Precision (AP) metrics is used, which is defined the area under Precision-Recall curve. It can be computed by integrating (or approximating the area under) the precision-recall curve with Equation A5:

$$AP = \sum_{n} (R_n - R_{n-1}) \times P_n \tag{A5}$$

where $P_n$ is the precision at the n-th threshold and $R-n$ is the recall at the n-th threshold. The mean Average Precision (mAP) is the mean of the AP scores across all classes in the dataset, providing an overall multi-class measure of model accuracy (Equation A6).

$$mAP = \frac{1}{C} \sum_{c=1}^{C} AP_c \tag{A6}$$

where $AP_c$ is the Average Precision for class c and C is the total number of classes, 4 in our dataset. The mAP can also be evaluated at different IoU thresholds (e.g., mAP@0.5 for IoU of 0.5 or mAP@[0.5:0.95] for an average over multiple IoU values from 0.5 to 0.95 in steps of 0.05).

## A2  YOLOv11-X Metrics

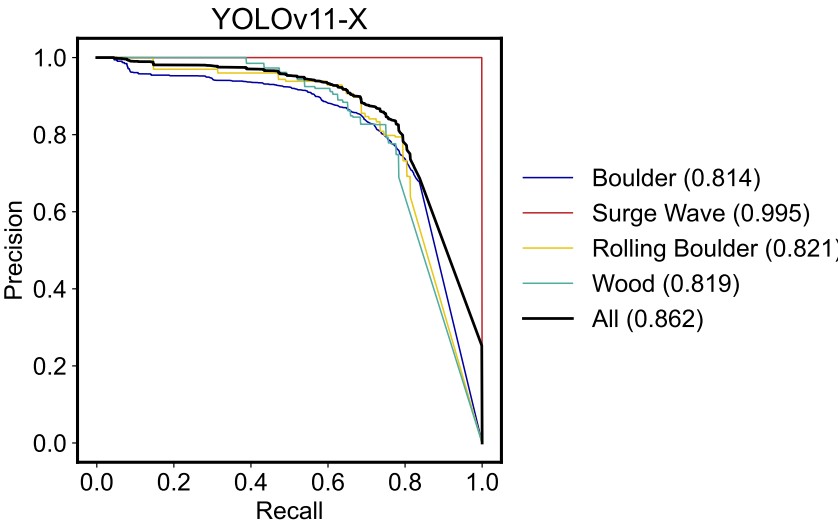

**Figure A1.** Precision-Recall (PR) curves for the YOLOv11-X performance on the test set across the object classes: Boulder, Surge Wave, Rolling Boulder and Wood (see also Tab. A1). Numbers in parentheses indicate the Average Precision (AP) for each class and the mean Average Precision (mAP) over all classes.

**Table A1.** Class-wise performance metrics of the YOLOv11-X model evaluated on the test dataset.

| Class | Instances | P | R | mAP50 | mAP50-95 |
|---|---|---|---|---|---|
| all | 3 476 | 0.817 | 0.823 | 0.862 | 0.583 |
| Boulder | 3 213 | 0.789 | 0.755 | 0.814 | 0.531 |
| Surge Wave | 9 | 0.9 | 1 | 0.995 | 0.762 |
| Rolling Boulder | 102 | 0.755 | 0.794 | 0.821 | 0.488 |
| Wood | 152 | 0.824 | 0.741 | 0.819 | 0.553 |

## A3 Object velocities

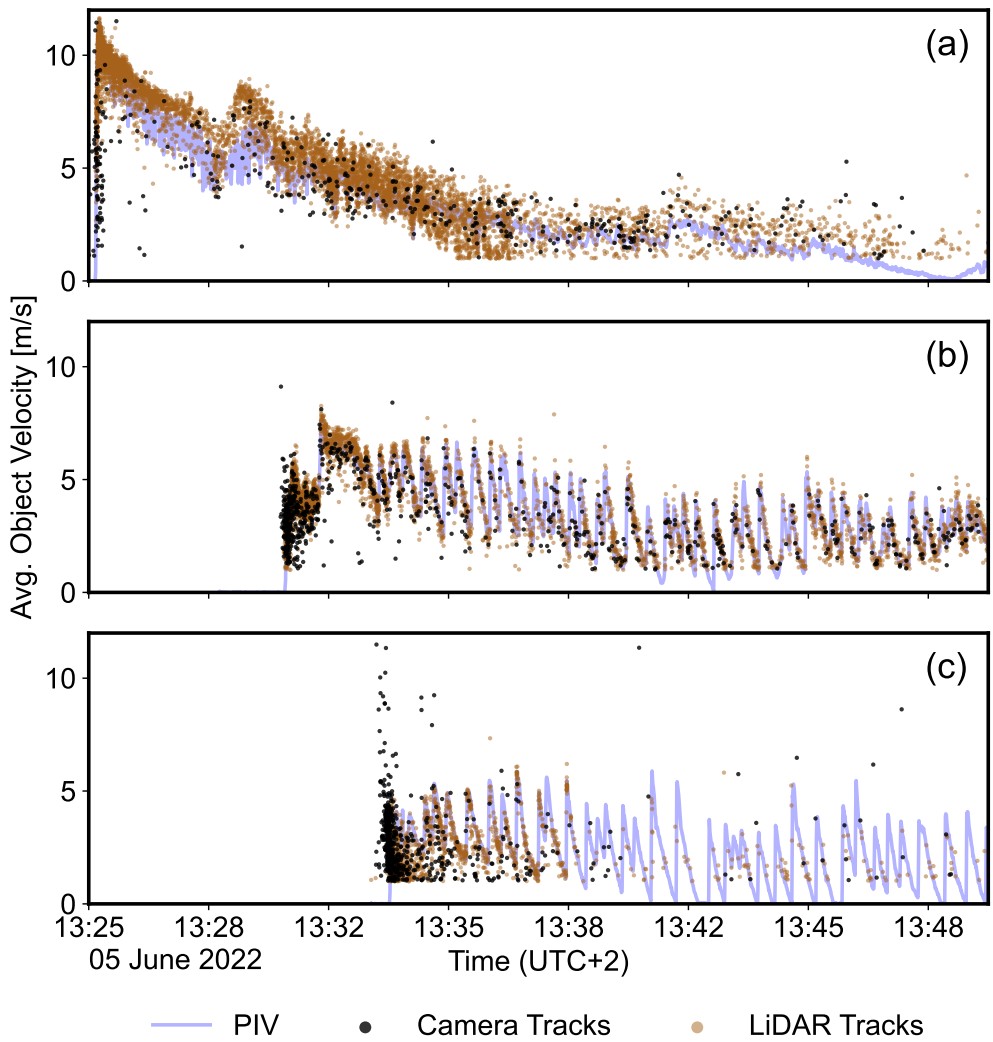

**Figure A2.** Mean object velocities calculated for the 5 June 2022 event, using the LiDAR hillshade and the camera LiDAR fusion method (Hirschberg et al., 2025) at the three monitoring stations: (a) Gazoduc (CDG), (b) CD27, and (c) CD29. Surface velocities derived from particle image velocimetry (PIV) by Spielmann et al. (2024) are included as a reference.

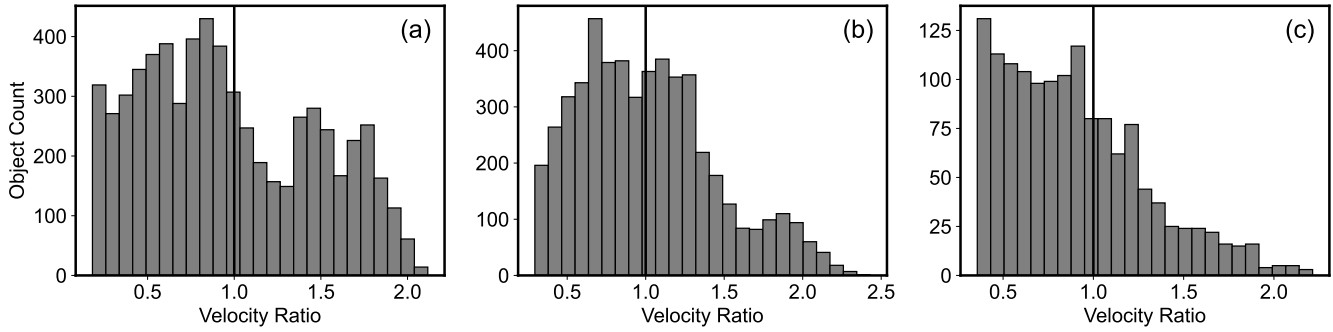

**Figure A3.** Distribution of object counts as a function of the velocity ratio for (a) CDG, (b) CD27, and (c) CD29. The velocity ratio is defined as $Velocity Ratio = Object Average Velocity / Flow Front Velocity$ for each object at each station. In CDG and CD27, a greater number of objects exhibit velocity ratios exceeding 1 compared to CD29, which also shows overall lower object counts. Note that this figure does not show temporal variations in object velocities relative to the flow front.

## A4  Grain sizes

**Table A2.** Particle sizes of the grain size distributions derived from LiDAR detections and object tracks for the 5 June 2022 event for stations CDG, CD27, and CD29. These particle sizes are described using various size metrics, such as D10, which represents the particle size at which 10% of the particles are finer and 90% are coarser.

| | Particle Size [m] | | | | |
|---|---|---|---|---|---|
| | $D_{10}$ | $D_{30}$ | $D_{50}$ | D60 | $D_{90}$ |
| Detections: | | | | | |
| CDG | 0.24 | 0.3 | 0.36 | 0.4 | 1.18 |
| CD27 | 0.24 | 0.3 | 0.39 | 0.46 | 1.15 |
| CD29 | 0.33 | 0.42 | 0.53 | 0.61 | 0.84 |
| Object Tracks: | | | | | |
| CDG | 0.28 | 0.34 | 0.4 | 0.45 | 1.25 |
| CD27 | 0.26 | 0.32 | 0.39 | 0.45 | 1.07 |
| CD29 | 0.35 | 0.41 | 0.46 | 0.5 | 0.76 |

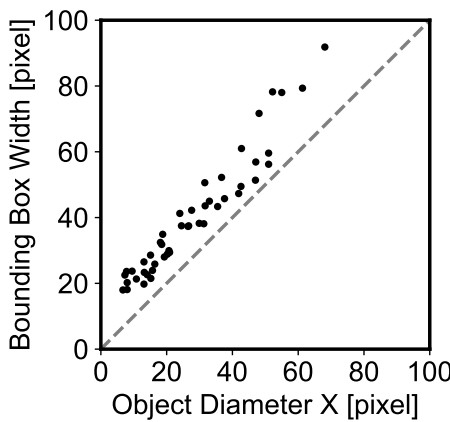

**Figure A4.** Comparison between object width and bounding box width (Y-axis) in pixels, demonstrating a near-linear relationship and highlighting the consistency between manually measured object diameters and bounding box predictions across the dataset. The observed offset of 12.2 pixels, corresponding to 24.4 cm, is likely attributed to the labeling approach, which accounts for turbulence and shadows surrounding the objects.

*Author contributions.* PES implemented the method, conducted the analysis, produced the figures and wrote the original article draft. PES, JH and JA designed the study. All authors contributed to the interpretation of the results and to the revision of the article.

*Competing interests.* The contact author has declared that neither they nor their co-authors have any competing interests.

*Acknowledgements.* The LiDAR installations were funded by the Swiss National Science Foundation (SNSF, grant number 193081) as well as funds from the Chair of Engineering Geology at ETH Zürich. We are grateful for technical support and scientific advice from Stefan Boss (WSL), Christoph Graf (WSL), Dr. Brian McArdell (WSL), Dr. Alexandre Badoux (WSL) and Amanda Åberg (WSL, ETH Zürich).

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
