# Peer review of "Deep learning-based object detection on LiDAR-derived hillshade images: Insights into grain size distribution and longitudinal sorting of debris flows"

_EGUsphere, 2025_

## Author Comment (AC1)

**egusphere-2025-743**

**Deep learning-based object detection on LiDAR-derived hillshade images: Insights into grain size distribution and longitudinal sorting of debris flows**

Paul E. Schmid, Jacob Hirschberg, Raffaele Spielmann, and Jordan Aaron

**Response to Referees**

We thank Pierluigi Confuorto very much for reviewing our article and for the constructive feedback. Below we provide our point-by-point responses.

1. It is my understanding that using hillshade representation, only 2 dimensions can be analyzed. Is there any possible improvement to obtain also the vertical dimension, which could be very important to be estimated?

Yes, your assumption is correct: we report 2D velocities and ignore the vertical component. Since the channel inclination on the fan is low  $(3-4^\circ)$ , the vertical component can be neglected. Fig. 6 supports this assumption, as the PIV velocities are in fact 3D (Aaron et al., 2025) and match well with the 2D object velocities. We will address this limitation and provide reasoning in the discussion. Additionally, we will note that 3D velocities could be obtained by projecting the vectors onto the 3D point cloud.

2. Would this methodology be implemented also to forecast trajectories of boulders and woods?

This is a good question, which we do not yet address in the article. As mentioned in the Introduction, we aim to develop a framework to efficiently obtain object trajectories. On the one hand, this will allow us to analyze object dynamics to better explain debris-flow phenomena such as longitudinal sorting and levee formation. On the other hand, if the model operates in real time, it could be used for early-warning purposes.

The method could also be used to test new numerical models that explicitly simulate the motion of large particles in the flow. This is a promising venue for future work, and we will add a paragraph to discuss the potential future use cases of our method.

3. Which are the error bounds of the different size materials in terms of velocities using Sort and Bot Sort?

Thank you for this comment. We did not clearly explain how we calculate object velocities based on bounding-box tracks. In the revision, we will specify that velocities are computed from the temporal displacement of the bounding-box centroid (x\_c, y\_c). Since we use the centroid rather than the box extent, the estimate is independent of box size. We will also add a comparison between manually annotated tracks and detector-derived centroids to quantify image-plane jitter ( $\sigma_x$ ,  $\sigma_y$  in pixels). The image-plane jitter represents the frame-to-frame variability in the detected bounding box position caused by minor localization noise.

4. Can this method be extended to differentiate submerged vs. surface-level boulders?

Thank you for this question. This is exactly why we differentiate between boulders and rolling boulders. According to our definition, rolling boulders are approximately the same size as the flow depth, meaning they have contact with the channel bed and are submerged. Boulders, on the other hand, are smaller and only visible because they are on the surface of the flow. We will clarify this distinction more effectively in the method section when defining the classes.

As a mere suggestion on the arrangement of the paper, I would split section 2 into 2.1 Geological and geomorphological setting (providing more info about the catchment area) and 2.2 about the monitoring set up. I find it a little bit confusing as it is.

All the best

Thank you for this suggestion. Since the geological and geomorphological context is not extensively described, we believe that a dedicated subsection is not necessary and could disrupt the flow of the text. However, we will merge Sections 2.1 and 2.2 and move the description of the monitoring setup to a new Section 2.2 in the revised manuscript.

**References:**

Aaron, J., Langham, J., Spielmann, R., Hirschberg, J., McArdell, B., Boss, S., Johnson, C. G., & Gray, J. M. N. T. (2025). Detailed observations reveal the genesis and dynamics of destructive debrisflow surges. *Communications Earth & Environment*, 6(1), 556. https://doi.org/10.1038/s43247-025-02488-7

---

## Author Comment (AC2)

**egusphere-2025-743**

**Deep learning-based object detection on LiDAR-derived hillshade images: Insights into grain size distribution and longitudinal sorting of debris flows**

Paul E. Schmid, Jacob Hirschberg, Raffaele Spielmann, and Jordan Aaron

**Response to Referees**

We thank the reviewer for taking the time to review our manuscript and for the constructive feedback. We provide our point-by-point responses below.

1. Please more explicitly describe the algorithms and parameters used for coordinate transformation, wobble correction, and artifact removal to ensure reproducibility.

Thank you for pointing out that this was not clear enough. We transformed the point-clouds into a coordinate system where the y-axis is parallel to the channel and the z-axis is vertical (cf. Spielmann & Aaron, 2024). Sensor wobble due to wind was corrected via a rigid iterative closest point (ICP) transformation, aligning successive scans by minimizing point-to-point distances (in stable areas outside the channel bed). This was only done for station CDG. Artifacts such as raindrop-induced holes were mitigated by filtering isolated returns and either locally interpolating or omitting the areas completely. We will expand the Methods section to list these steps and parameters explicitly.

2. Authors should briefly explain why these six specific models (YOLO variants, RT-DETR, etc.) were chosen over others, especially given their similar performance.

We selected models to represent both one-stage (YOLO series, RetinaNet) and two-stage (Faster R-CNN) detectors, plus a recent Transformer-based detector (RT-DETR). Together, these cover the main approaches of modern object detection (e.g., Chen et al., 2024; Zhao et al., 2024; Zou et al., 2023). Within YOLO, we compared three recent generations (v5, v8, v11) to assess whether architectural improvements affect debris-flow feature detection. For example, small objects are generally harder to detect, and sensitivity can differ across architectures. Including multiple architectures helps ensure our results are not biased by a single detection paradigm. We will better highlight in the introduction that these are the most widely used object detection models and represent different architectural classes of models.

3. Could you discuss why BoT-SORT was ultimately preferred over SORT in more detail, especially regarding high-velocity object handling?

Thank you for pointing out that this was not clear. SORT failed for many high-velocity objects (>5 m s-1), which are common in debris flows. BoT-SORT addresses this by (i) expanding the Kalman-filter state for improved localization and (ii) compensating for global motion (Aharon et al., 2022). In practice, this yielded more stable track IDs and longer trajectories for fast-moving boulders. We will clarify this in the discussion of our

results and add a figure showing a SORT and BoT-SORT example to the Supporting Information.

4. Add confidence intervals or p-values to velocity and size comparisons to strengthen quantitative claims.

Thank you for this suggestion. We agree that confidence interval estimates will strengthen our quantitative claims. In the revision, we will report confidence intervals and provide a discussion of uncertainty, similar to the comment given to Referee 1. Furthermore, we will include a comparison of velocities and object sizes derived from manually labeled tracks and detector-based tracks to clarify the uncertainty estimation.

5. How is the loss of 3D information in hillshade projection? And its effect on vertical velocity underestimation?

Thank you for bringing up this point. We report 2D velocities and ignore the vertical component. The channel inclination on the fan is low (3-4°), so the vertical component of velocity is relatively minor. Fig. 6 supports this assumption, as the PIV velocities are in fact 3D and match well with the 2D object velocities. We will mention this limitation and reasoning in the discussion, and also that 3D velocities could be obtained by projecting the vectors to the 3D point cloud.

**References:**

Aharon, N., Orfaig, R., & Bobrovsky, B.-Z. (2022). *BoT-SORT: Robust Associations Multi-Pedestrian Tracking* (No. arXiv:2206.14651). arXiv. http://arxiv.org/abs/2206.14651

Chen, W., Luo, J., Zhang, F., & Tian, Z. (2024). A review of object detection: Datasets, performance evaluation, architecture, applications and current trends. *Multimedia Tools and Applications*, 83(24), 65603–65661. https://doi.org/10.1007/s11042-023-17949-4

Spielmann, R., & Aaron, J. (2024). A new method for detailed discharge and volume measurements of debris flows based on high-frequency 3D LiDAR point clouds; Illgraben, Switzerland. *Engineering Geology*, 329, 107386. https://doi.org/10.1016/j.enggeo.2023.107386

Zhao, Y., Lv, W., Xu, S., Wei, J., Wang, G., Dang, Q., Liu, Y., & Chen, J. (2024). *DETRs Beat YOLOs on Real-time Object Detection* (No. arXiv:2304.08069). arXiv. http://arxiv.org/abs/2304.08069

Zou, Z., Chen, K., Shi, Z., Guo, Y., & Ye, J. (2023). Object Detection in 20 Years: A Survey. *Proceedings of the IEEE*, 111(3), 257–276. https://doi.org/10.1109/JPROC.2023.3238524

---

## Author Response (AR1)

**Response to Reviewers**

**EGUSPHERE-2025-743**

**Deep learning-based object detection on LiDAR-derived hillshade images: Insights into grain size distribution and longitudinal sorting of debris flows**

Paul Emil Schmid[1], Jacob Hirschberg[1,2], Raffaele Spielmann[1,2], and Jordan Aaron[1,2]

[1]Chair of Engineering Geology, Department of Earth and Planetary Sciences, ETH Zürich
[2]Swiss Federal Institute for Forest, Snow, and Landscape Research WSL

November 18, 2025

We thank Pierluigi Confuorto and the anonymous referee for reviewing our article and for providing constructive feedback. We believe that the reviewer comments helped to improve the clarity and quality of the manuscript. In the following sections, we present all reviewer comments (in blue) together with our responses (in black). Where appropriate, excerpts from the updated text are indented to illustrate the revisions made.

**Referee 1**

1. It is my understanding that using hillshade representation, only 2 dimensions can be analyzed. Is there any possible improvement to obtain also the vertical dimension, which could be very important to be estimated?

   Yes, your assumption is correct: we report two-dimensional (2D) velocities and neglect the vertical component. Given the low channel inclination on the fan (3-4°), the vertical contribution can reasonably be ignored. Figure 6 supports this assumption, as the PIV-derived velocities are three-dimensional (Aaron et al., 2025) and correspond well with the 2D object velocities. The loss of vertical velocity information resulting from the 2D hillshade visualization is now discussed in the revised manuscript (Lines 376-382).

   Lines 376-382: *Another limitation arises from the conversion of 3D LiDAR point clouds into 2D hillshade images, which leads to the loss of spatial information along the vertical axis. As a result, the computed object velocities are constrained to a 2D plane, despite the fact that boulders may submerge and re-emerge, indicating vertical velocity components that remain uncaptured. However, the impact of this limitation is expected to be small, as the vertical velocity component is minor compared to the horizontal one. This is due to the low channel inclination on the fan (3–4°), and the 2D object velocities correspond well with the PIV-derived velocities (cf. Fig. 6/A2), which represent full 3D measurements (Aaron et al., 2025). 3D velocities could, however, be calculated by projecting the bounding box centroids on the 3D point clouds.*

2. Would this methodology be implemented also to forecast trajectories of boulders and woods?

   As mentioned in the Introduction (Lines 62-69), our goal is to develop a framework for efficiently extracting object trajectories. This will enable the analysis of object dynamics to better explain debris-flow processes such as longitudinal sorting and levee formation. Moreover, if implemented in real time, the approach could contribute to early-warning applications. The method may also support the evaluation of numerical models that explicitly simulate the motion of large particles, with object trajectories from past events serving as valuable validation data. At present, the approach primarily functions as a monitoring workflow, but it represents a promising direction for future research.

   Lines 62-69: *In this study, we introduce a new method to detect and track debris-flow features based solely on LiDAR data, thereby being independent of ambient light, and enabling the estimation of object velocities, sizes, and trajectories for thousands of debris-flow features. We analyze data captured by multiple high-resolution LiDAR sensors installed at various locations along the channel of a Swiss debris-flow catchment (Illgraben) during a debris-flow event, allowing us to study debris-flow evolution over time and space. Visualizing point clouds in hillshades simplified the processing and allowed us to utilize modern CNN object detection algorithms, originally developed for optical images. Here, we analyze and compare three versions of YOLO (v5, v8,*

*and v11), along with RT-DETR, RetinaNet, and Faster-RCNN for detection and tracking of debris-flow features. This automated approach establishes a foundation for future investigations and early-warning of debris-flow events across diverse locations and conditions.*

3. Which are the error bounds of the different size materials in terms of velocities using Sort and Bot Sort?

Thank you for this comment. Two primary error sources are acknowledged in the computation of object velocity. The first relates to variations in bounding-box size, which could potentially influence velocity estimates. However, this source of error can be mitigated by averaging velocity over the entire object track and utilizing the bounding-box centroid rather than the box edges. In the revised manuscript, it has been clarified that object velocities are computed from the temporal displacement of the bounding-box centroid ($x_c$, $y_c$), as described in Lines 227-229 and Equation 1. The use of the centroid rather than the box extent ensures that velocity estimates remain independent of variations in bounding-box size.

Lines 227-229: *First, we converted the bounding box coordinates from pixel units to meters using the hillshade image resolution of 0.02 m/px. The (instantaneous) velocity was then calculated based on the temporal displacement of the bounding box centroid between consecutive frames (Eq. 2).*

Equation 1:

$$v(t_i) = \frac{\sqrt{(x_c(t_i) - x_c(t_{i-1}))^2 + (y_c(t_i) - y_c(t_{i-1}))^2}}{t_i - t_{i-1}} \tag{1}$$

where $v_{t_i}$ is the velocity of a tracked debris-flow feature and $x_c$ and $y_c$ are the centroids object coordinates at time t . The time step between two frames is 0.1 s, corresponding to the LiDAR recording frequency of 10 Hz.

The second error source concerns the quality of Multi-Object Tracking (MOT). Validation of MOT results is particularly challenging, as objects can be occluded or re-identified, leading to class switches or fragmented trajectories (Luo et al., 2021; Milan et al., 2013). Result validation is further complicated by: (1) variability in performance metrics (e.g., MOTA, IDF1, HOTA), (2) the lack of standardized ground truth across different environments, and (3) reproducibility issues in existing benchmarks (Milan et al., 2013). Furthermore, the present method does not track persons or vehicles, for which established benchmarks such as Milan et al., 2016 exist. Consequently, object velocities are validated against the PIV (Particle Image Velocimetry) velocities presented by Spielmann et al., 2024 and validated by those authors. In lines 366-371 of the revised manuscript we clarify this limitation.

Lines 366-371: *Another uncertainty in the computation of object velocities concerns the quality of MOT, where validation is particularly challenging due to object occlusion and re-identification issues that lead to class switches or fragmented trajectories (Luo et al., 2021). The variability in performance metrics (e.g., MOTA, IDF1, HOTA), lack of standardized ground truth, and reproducibility issues are well-known challenges in existing benchmarks (Milan et al., 2013). Furthermore, established MOT benchmarks such as (Milan et al., 2016) focus on persons or vehicles rather than debris-flow objects. Therefore, the presented object velocities were validated against the PIV method as mentioned above.*

4. Can this method be extended to differentiate submerged vs. surface-level boulders?

Thank you for this question. This distinction is precisely the reason why we differentiate between boulders and rolling boulders. According to our definition, rolling boulders are approximately the same size as the flow depth, maintaining contact with the channel bed and remaining partially submerged. Boulders, in contrast, are smaller and visible primarily because they are transported at the flow surface (cf. Aaron et al., 2023). This clarification has been incorporated in lines 135-141 of the revised manuscript.

Lines 135-141: *The objects of interest were defined and tested in previous studies by Aaron et al., 2023; Hirschberg et al., 2025; Spielmann and Aaron, 2024, and these definitions were adopted in this work. The four feature classes include Boulders, Surge Waves, Rolling Boulders, and Woody Debris (Fig. 4). These object types are hypothesized to represent distinct vertical velocity profiles within the debris flow. Woody debris, which floats on the surface, is assumed to travel at the surface velocity. In contrast, rolling boulders, which are approximately of the same size as the flow depth and remain in contact with the channel bed, rather represent the depth-averaged velocity (Aaron et al., 2023).*

As a mere suggestion on the arrangement of the paper, I would split section 2 into 2.1 Geological and geomorphological setting (providing more info about the catchment area) and 2.2 about the monitoring set up. I find it a little bit confusing as it is.

Thank you for this suggestion. We have split Sections 2.1 and 2.2, with the revised Section 2.1 now describing the Illgraben catchment system. Section 2.2 provides an overview of the monitoring setup and the observed debris-flow event. In addition, we have added a paragraph summarizing the geological setting in Section 2.1 from line 77-84 of the revised manuscript.

Lines 77-84: *The catchment is underlain by fractured bedrock forming an anticline within the Penninic nappe stack (Schlunegger et al., 2009). The Illgraben channel follows the southwest-trending axial plane of this structure, incising Triassic schists and dolobreccias that provide the fine-grained matrix of the debris-flow deposits. Steep limestone cliffs ($>50°$) on the northwest side primarily generate rockfalls, whereas quartzite slopes on the southeast side ($30–40°$) produce frequent debris flows that supply quartzite clasts and fine material to the channel (Schlunegger et al., 2009). Further information on the geological and geomorphological setting of the Illgraben catchment is provided, for example, by Gabus et al., 2008; Schlunegger et al., 2009 and McArdell and Sartori, 2021.*

**Referee 2**

1. Please more explicitly describe the algorithms and parameters used for coordinate transformation, wobble correction, and artifact removal to ensure reproducibility.

Thank you for pointing out that this was not clear enough. We transformed the point-clouds into a coordinate system where the y-axis is parallel to the channel and the z-axis is vertical (cf. Spielmann and Aaron, 2024). Sensor wobble due to wind was corrected via a rigid iterative closest point (ICP) transformation, aligning successive scans by minimizing point-to-point distances (in stable areas outside the channel bed). This was only done for station CDG. Artifacts such as raindrop-induced holes were mitigated by filtering isolated returns and either locally interpolating or omitting the areas completely. This explanation was added to the revised manuscript in lines 115-121.

Lines 115-121: *Corrections were applied to account for sensor wobbling caused by wind impacts on the LiDAR installation by performing a rigid iterative closest point (ICP) transformation that aligned successive scans through minimization of point-to-point distances in stable areas outside the channel bed. This correction was applied only to station CDG. Artifacts in the point clouds, such as holes resulting from raindrops on the LiDAR sensor, were removed by selecting a specific area of the point cloud. The point clouds were further cropped to include only the section of the channel upstream of the check dam (Fig. 3b), where the resolution is highest and the results remain unaffected by the presence of the check dam. Additionally, to ensure consistent image dimensions, all point clouds were cropped using the same X and Y limits.*

2. Authors should briefly explain why these six specific models (YOLO variants, RT-DETR, etc.) were chosen over others, especially given their similar performance.

We selected models to represent both one-stage (YOLO series, RetinaNet) and two-stage (Faster R-CNN) detectors, plus a recent Transformer-based detector (RT-DETR). Together, these cover the main approaches of modern object detection (e.g., Chen et al., 2024; Zhao et al., 2024; Zou et al., 2023). Within YOLO, we compared three recent generations (v5, v8, v11) to assess whether architectural improvements affect debris-flow feature detection. For example, small objects are generally harder to detect, and sensitivity can differ across architectures (Li et al., 2020). Including multiple architectures helps ensure our results are not biased by a single detection paradigm. In the revised manuscript (Lines 147-151), we added a comment discussing the comparison of model architectures with respect to their ability to detect small and partially occluded objects.

Lines 147-151: *We tested and compared four state-of-the-art object detection models, including three versions of YOLO (v5, v8, and v11), as well as RT-DETR, RetinaNet, and Faster R-CNN, using our custom hillshade dataset. The models were selected based on their popularity and performance, all ranking among the top performers in the COCO benchmark (cf. Zou et al., 2023). In addition, we evaluated multiple architectures to assess their ability to accurately detect small and partially occluded objects, as model performance can vary substantially across object scales (Zou et al., 2023).*

3. Could you discuss why BoT-SORT was ultimately preferred over SORT in more detail, especially regarding high-velocity object handling?

Thank you for pointing out that this was not clear. SORT failed for many high-velocity objects ($> 5\text{ms}^{-1}$), which are common in debris flows. BoT-SORT addresses this by (i) expanding the Kalman-filter state for improved localization and (ii) compensating for global motion (Aharon et al., 2022). In practice, this yielded more stable track IDs and longer trajectories for fast-moving boulders. This is described in Section 3.3.3 of the manuscript.

Section 3.3.3: *The Simple Online and Realtime Tracking (SORT) algorithm (Bewley, 2024; Bewley et al., 2016) predicts the movement of detected objects between frames using a linear constant velocity model and refines these predictions with a Kalman Filter. To associate predicted positions with actual detections, SORT utilizes the Hungarian algorithm, optimizing the assignment based on an IoU metric. Each object is assigned a unique identifier, which is maintained across frames if the association is successfully established. However, preliminary testing showed that the SORT algorithm performed poorly for objects with velocities greater than 5 m/s. To overcome this limitation, we applied the BoT-SORT algorithm (Aharon et al., 2022). BoT-SORT builds on the tracking-by-detection approach used in SORT but incorporates an improved Kalman filter state vector for more accurate box localization and a camera motion compensation feature.*

4. Add confidence intervals or p-values to velocity and size comparisons to strengthen quantitative claims.

Thank you for this comment. Two primary error sources are acknowledged in the computation of object velocity. The first relates to variations in bounding-box size, which could potentially influence velocity estimates. However, this source of error can be mitigated by averaging velocity over the entire object track and utilizing the bounding-box centroid rather than the box edges. In the revised manuscript, it has been clarified that object velocities are computed from the temporal displacement of the bounding-box centroid ($x_c$, $y_c$), as described

in Lines 227-229 and Equation 1. The use of the centroid rather than the box extent ensures that velocity estimates remain independent of variations in bounding-box size.

Lines 227-229: *First, we converted the bounding box coordinates from pixel units to meters using the hill-shade image resolution of 0.02 m/px. The (instantaneous) velocity was then calculated based on the temporal displacement of the bounding box centroid between consecutive frames (Eq. 2).*

Equation 1:

$$v(t_i) = \frac{\sqrt{(x_c(t_i) - x_c(t_{i-1}))^2 + (y_c(t_i) - y_c(t_{i-1}))^2}}{t_i - t_{i-1}} \tag{2}$$

where $v_{t_i}$ is the velocity of a tracked debris-flow feature and $x_c$ and $y_c$ are the centroids object coordinates at time t . The time step between two frames is 0.1 s, corresponding to the LiDAR recording frequency of 10 Hz.

The second error source concerns the quality of Multi-Object Tracking (MOT). Validation of MOT results is particularly challenging, as objects can be occluded or re-identified, leading to class switches or fragmented trajectories (Luo et al., 2021; Milan et al., 2013). Result validation is further complicated by: (1) variability in performance metrics (e.g., MOTA, IDF1, HOTA), (2) the lack of standardized ground truth across different environments, and (3) reproducibility issues in existing benchmarks (Milan et al., 2013). Furthermore, the present method does not track persons or vehicles, for which established benchmarks such as Milan et al., 2016 exist. Consequently, object velocities are validated against the PIV (Particle Image Velocimetry) velocities presented by Spielmann et al., 2024 and validated by those authors. In lines 366-371 of the revised manuscript we clarify this limitation.

Lines 366-371: *Another uncertainty in the computation of object velocities concerns the quality of MOT, where validation is particularly challenging due to object occlusion and re-identification issues that lead to class switches or fragmented trajectories (Luo et al., 2021). The variability in performance metrics (e.g., MOTA, IDF1, HOTA), lack of standardized ground truth, and reproducibility issues are well-known challenges in existing benchmarks (Milan et al., 2013). Furthermore, established MOT benchmarks such as (Milan et al., 2016) focus on persons or vehicles rather than debris-flow objects. Therefore, the presented object velocities were validated against the PIV method as mentioned above.*

Thank you for suggesting testing hypotheses with p-values. We agree that this could strengthen claims. However, we use the distributions merely for visualization and qualitative analysis. At this point, we don't think statistical tests are meaningful due to the different number of detections through time and across at the stations and the mentioned differences in senor resolutions (CD27).

5. How is the loss of 3D information in hillshade projection? And its effect on vertical velocity underestimation?

Thank you for bringing up this point. We report two-dimensional (2D) velocities and neglect the vertical component. Given the low channel inclination on the fan (3-4°), the vertical contribution can reasonably be ignored. Figure 6 supports this assumption, as the PIV-derived velocities are three-dimensional (Aaron et al., 2025) and correspond well with the 2D object velocities. The loss of vertical velocity information resulting from the 2D hillshade visualization is now discussed in the revised manuscript (Lines 376-382).

Lines 376-382: *Another limitation arises from the conversion of 3D LiDAR point clouds into 2D hillshade images, which leads to the loss of spatial information along the vertical axis. As a result, the computed object velocities are constrained to a 2D plane, despite the fact that boulders may submerge and reemerge, indicating vertical velocity components that remain uncaptured. However, the impact of this limitation is expected to be small, as the vertical velocity component is minor compared to the horizontal one. This is due to the low channel inclination on the fan (3–4°), and the 2D object velocities correspond well with the PIV-derived velocities (cf. Fig. 6/A2), which represent full 3D measurements (Aaron et al., 2025). 3D velocities could, however, be calculated by projecting the bounding box centroids in the 3D point clouds to compute three-dimensional velocities.*

**Technical Revisions**

Technical revisions (in blue) are listed below, together with comments and references to their respective locations in the manuscript (in black).

- Figure 1b and 1c: Please indicate view direction in the caption (or on the photo).

  A note was added to the caption of Figure 1b and Figure 1c. The photograph in Figure 1b was taken from the channel *facing downstream*, while the photograph in Figure 1c was taken *facing upstream*. The inserted text appears in line 5 of the Figure 1 caption.

- Figure 3a: Please add a tentative scale and view direction of the photo.

  A scale bar indicating approximately 4 m (corresponding to the left half of the channel) was added to Figure 3a (bottom left).

- Figure 4: Please add scale bars to the hillshade excerpts.

  Scale bars were added to Figure 4a, 4b, and 4c below the graphic, ranging from 0 to 3 m. Figure 4d shows the entire channel width, for which a 10 m scale bar was added.

- Short summary: Please revise to meet the length and abbreviation requirements.

  An abbreviation was removed, and the summary was shortened to meet the 500-character limit.

- Figure 6: Please ensure accessibility for readers with colour vision deficiencies.

  The colormap of Figure 6 was adjusted to improve visibility for readers with colour vision deficiencies.

**References**

Aaron, J., Langham, J., Spielmann, R., Hirschberg, J., McArdell, B., Boss, S., Johnson, C. G., & Gray, J. M. N. T. (2025). Detailed observations reveal the genesis and dynamics of destructive debris-flow surges. *Communications Earth & Environment*, *6*(1), 556. https://doi.org/10.1038/s43247-025-02488-7

Aaron, J., Spielmann, R., McArdell, B. W., & Graf, C. (2023). High-Frequency 3D LiDAR Measurements of a Debris Flow: A Novel Method to Investigate the Dynamics of Full-Scale Events in the Field. *Geophysical Research Letters*, *50*(5), e2022GL102373. https://doi.org/10.1029/2022GL102373

Aharon, N., Orfaig, R., & Bobrovsky, B.-Z. (2022, July). BoT-SORT: Robust Associations Multi-Pedestrian Tracking. Retrieved November 19, 2024, from http://arxiv.org/abs/2206.14651

Bewley, A. (2024). Abewley/sort. Retrieved October 3, 2024, from https://github.com/abewley/sort

Bewley, A., Ge, Z., Ott, L., Ramos, F., & Upcroft, B. (2016). Simple online and realtime tracking. *2016 IEEE International Conference on Image Processing (ICIP)*, 3464–3468. https://doi.org/10.1109/ICIP.2016.7533003

Chen, W., Luo, J., Zhang, F., & Tian, Z. (2024). A review of object detection: Datasets, performance evaluation, architecture, applications and current trends. *Multimedia Tools and Applications*, *83*(24), 65603–65661. https://doi.org/10.1007/s11042-023-17949-4

Gabus, J. H., Weidmann, M., Bugnon, P.-C., Burri, M., Sartori, M., & Marthaler, M. (2008). Geological map of Sierre, LK 1278, sheet 111.

Hirschberg, J., Tertius Bickel, V., & Aaron, J. (2025). Deep-Learning-Based Object Detection and Tracking of Debris Flows in 3-D Through LiDAR-Camera Fusion. *IEEE Transactions on Geoscience and Remote Sensing*, *63*, 1–13. https://doi.org/10.1109/TGRS.2025.3609573

Li, Y., Li, S., Du, H., Chen, L., Zhang, D., & Li, Y. (2020). YOLO-ACN: Focusing on Small Target and Occluded Object Detection. *IEEE Access*, *8*, 227288–227303. https://doi.org/10.1109/ACCESS.2020.3046515

Luo, W., Xing, J., Milan, A., Zhang, X., Liu, W., & Kim, T.-K. (2021). Multiple object tracking: A literature review. *Artificial Intelligence*, *293*, 103448. https://doi.org/10.1016/j.artint.2020.103448

McArdell, B. W., & Sartori, M. (2021). The Illgraben Torrent System. In E. Reynard (Ed.), *Landscapes and Landforms of Switzerland* (pp. 367–378). Springer International Publishing. https://doi.org/10.1007/978-3-030-43203-4_25

Milan, A., Leal-Taixe, L., Reid, I., Roth, S., & Schindler, K. (2016). MOT16: A Benchmark for Multi-Object Tracking. https://doi.org/10.48550/ARXIV.1603.00831

Milan, A., Schindler, K., & Roth, S. (2013). Challenges of Ground Truth Evaluation of Multi-target Tracking. *2013 IEEE Conference on Computer Vision and Pattern Recognition Workshops*, 735–742. https://doi.org/10.1109/CVPRW.2013.111

Schlunegger, F., Badoux, A., McArdell, B. W., Gwerder, C., Schnydrig, D., Rieke-Zapp, D., & Molnar, P. (2009). Limits of sediment transfer in an alpine debris-flow catchment, Illgraben, Switzerland. *Quaternary Science Reviews*, *28*(11-12), 1097–1105. https://doi.org/10.1016/j.quascirev.2008.10.025

Spielmann, R., & Aaron, J. (2024). A new method for detailed discharge and volume measurements of debris flows based on high-frequency 3D LiDAR point clouds; Illgraben, Switzerland. *Engineering Geology*, *329*, 107386. https://doi.org/10.1016/j.enggeo.2023.107386

Spielmann, R., Huber, S., & Aaron, J. (2024). Direct measurements of debris-flow feature velocities using high-frequency 3D LiDAR scanners. *XIVth International Symposium on Landslides*.

Zhao, Y., Lv, W., Xu, S., Wei, J., Wang, G., Dang, Q., Liu, Y., & Chen, J. (2024, April). DETRs Beat YOLOs on Real-time Object Detection. Retrieved November 4, 2024, from http://arxiv.org/abs/2304.08069

Zou, Z., Chen, K., Shi, Z., Guo, Y., & Ye, J. (2023). Object Detection in 20 Years: A Survey. *Proceedings of the IEEE*, *111*(3), 257–276. https://doi.org/10.1109/JPROC.2023.3238524